

# Improved method of estimating temperatures at meteor peak heights

Emranul Sarkar[1,2], Alexander Kozlovsky[1], Thomas Ulich[1], Ilkka Virtanen[2], Mark Lester[3], and Bernd Kaifler[4]

[1]Sodankylä Geophysical Observatory, Sodankylä, Finland
[2]Space Physics and Astronomy Research Unit, University of Oulu, Finland
[3]Department of Physics and Astronomy, University of Leicester, Leicester, UK
[4]Deutsches Zentrum für Luft- und Raumfahrt, Institut für Physik der Atmosphäre, Oberpfaffenhofen, Germany

**Correspondence:** Emranul Sarkar (emranul.sarkar@oulu.fi)

**Abstract.**

For two decades meteor radars have been routinely used to monitor temperatures around the 90 km altitude. A common method, based on a temperature-gradient model, is to use the height dependence of meteor decay time to obtain a height-averaged temperature in the peak meteor region. Traditionally this is done by fitting a linear regression model in the scattered

plot of $log_{10}(1/\tau)$ and height, where $\tau$ is the half-amplitude decay time of the received signal. However, this method was found to be consistently biasing the slope estimate. The consequence of such bias is that it produces a systematic offset in the estimated temperature, and thus requiring calibration with other colocated measurements. The main reason for such a biasing effect is thought to be due to the failure of the classical regression model to take into account the measurement error in $\tau$ or the observed height. This is further complicated by the presence of various geophysical effects in the data, which are not taken

into account in the physical model. The effect of such biasing is discussed on both theoretical and experimental grounds. An alternative regression method that incorporates various error terms in the statistical model is used for line fitting. This model is used to construct an analytic solution for the bias-corrected slope coefficient for this data. With this solution, meteor radar temperatures can be obtained independently without using any external calibration procedure. When compared with colocated lidar measurements, the temperature estimated using this method is found to be accurate within 7% or better and without any

systematic offset.

## 1   Introduction and background

As meteoroids enter the Earth's atmosphere, they produce ionized trails which can be detected as back-scattered radio signals by interferometric radars. After the trail has been formed, the ionisation begins to dissipate by various processes, such as,

ambipolar diffusion, eddy diffusion, or electron loss due to recombination and attachment depending on the height of ablation.





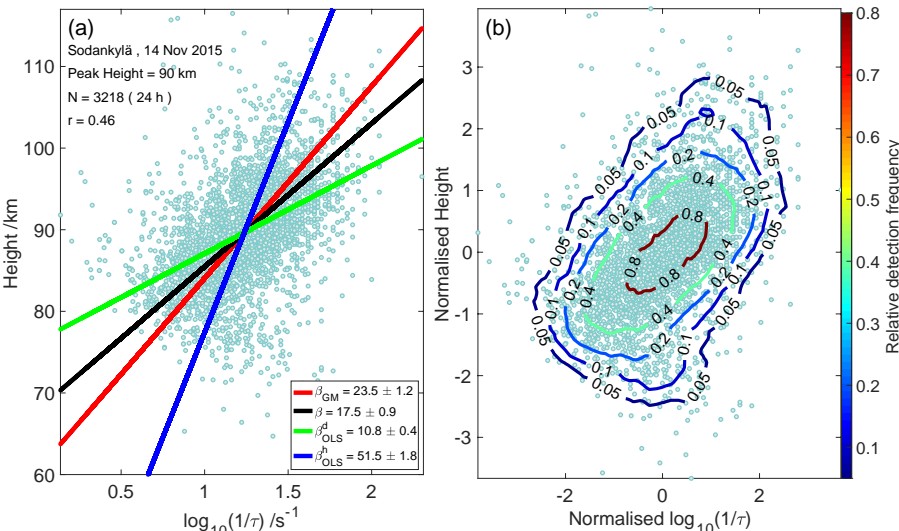

**Figure 1.** (a) Typical scatter plot of $log_{10}(1/\tau)$ and height. The lines correspond to best fit models using different regression methods described in the text. The *green* and *blue* line corresponds to 'ordinary least-squares method (OLS)' with $log_{10}(1/\tau)$ and height as independent variable respectively. The *red* line correspond to the geometric mean (GM) of $\beta^d_{OLS}$ and $\beta^h_{OLS}$. The *black* line is the bias-corrected slope obtained by using Eq. (37). (b) The bivariate distribution of the data. The measured height and $log_{10}(1/\tau)$ are converted to dimension free coordinates using Eq. (18). The relative density contours are obtained by counting the number of detections in a radius of 0.5 relative to the density at the height of peak meteor occurrences at the center.

The rate at which the echo power decreases is also determined by the combined effect of electron line density of the trail, ambient pressure and temperature.

If the electron line density of the trail is less than $2.4 \times 10^{14}$ electrons $\mathrm{m}^{-1}$ the trail is called 'underdense', meaning each electron in the trail scatters independently. The decay of underdense trails is thought to be mainly due to ambipolar diffusion at a
height range of 85–95 km, where the majority of the meteors ablate (Jones, 1975). In the weak scattering limit the backscattered amplitude of the radio signal from an underdense trail decays with time ($t$) as

$$A(t) = A(0)e^{-16\pi^2 D_a t/\lambda_r^2} \tag{1}$$

where $\lambda_r$ is the radar wavelength and $D_a$ is the ambipolar diffusion coefficient (Kaiser, 1953). This coefficient depends on the ambient pressure ($P$) and temperature ($T$) of the neutral gas (Chilson et al., 1996) and can be estimated from the half-amplitude
decay time ($\tau$) as

$$D_a = K_{amb}\frac{T^2}{P} = \frac{\lambda_r^2 \ln 2}{16\pi^2 \tau} \tag{2}$$

$K_{amb}$ in Eq. (2) is a constant related to the ionic constituent of the plasma in the trail (Hocking et al., 1997). The pressure at a given height ($h$) is

$$P(h) = P(0)e^{-\int_0^h \frac{mg}{kT(z)}dz} \tag{3}$$



where $m$ is the mass of a typical atmospheric molecule, $g$ is the acceleration of gravity, $k$ is the Boltzmann constant, $z$ is an axis along the vertical. Substituting the equation for pressure in Eq. (2), and differentiating provides the height profile of the decay time:

$$log_{10} D_a(h) = 2 log_{10} T(h) + log_{10} e \frac{mg}{k} \int_0^h \frac{1}{T(z)} dz + \Psi \tag{4}$$

$$\frac{d}{dh} log_{10}\left(\frac{1}{\tau(h)}\right) = 2\, log_{10} e \frac{dT}{dh}\frac{1}{T(h)} + log_{10} e \frac{mg}{k}\frac{1}{T(h)} \tag{5}$$

where $\Psi$ is a constant. Equation (5) states that the height profile of decay time is a function of both temperature and temperature gradient under the assumption of ambipolar diffusion for underdense meteor trails. In practice, most trail echoes are received at a small altitude range referred to as the region of peak meteor occurrence (Hocking, 1999). Hence a height-averaged temperature gradient near the peak height can be used to estimate the mean temperature ($T_0$) at the peak height by fitting a linear function (Hocking, 1999). A linear approximation of Eq. (5) is,

$$T_0 = \beta \left( 2 < \frac{dT}{dh} > + \frac{mg}{k} \right) log_{10} e \tag{6}$$

where $\beta$ is the slope of the scattered plot of $log_{10}(1/\tau)$ and height, and $T_0$ is the average temperature of the atmosphere at the height of peak meteor occurrence. A typical scattered plot of height and $log_{10}(1/\tau)$ shows significant variation in the measured data along both abscissa and ordinate (Fig. 1). Traditionally, the slope ($\beta$) is estimated using the ordinary least-squares (OLS) method with $log_{10}(1/\tau)$ as the independent variable. The justification of using $log_{10}(1/\tau)$ as independent

variable is that the measurement errors in $\tau$ are smaller than those in heights (Hocking et al., 1997). While the pulse length, angular resolution etc., of the radar introduces intrinsic measurement errors in heights, much of the variation in decay time is due to various geophysical effects that persist at all altitudes. At higher altitude the collision frequency with neutrals is reduced, and the diffusion is inhibited in a direction orthogonal to the geomagnetic field (Jones, 1991; Robson, 2001). This anisotropic diffusion causes an increase in the duration of meteor radar echoes as compared to ambipolar diffusion. Whereas at lower

altitude decay time tends to decrease due to additional effect of electron loss by recombination and attachment (Younger et al., 2008). In addition, other geophysical factors, such as meteor fragmentation, turbulence within trail, chemical composition of the meteors, or the temperature variation due to passage of tides and gravity waves, can contribute to the measurements of decay time and heights at all altitudes (Hocking, 2004).

Temperature estimation from meteor radar (MR) data requires obtaining the best-fit regression line in the scattered plot of

$log_{10}(1/\tau)$ and height. However, the pioneer work done by Hocking (1999) to implement this method using ordinary least-squares fitting showed a clear systematic offset between the MR temperature and colocated lidar measurements, indicating that the estimated slope was not determined correctly. To correct for this offset, a common practice is to calibrate the meteor radar temperatures using temperatures from lidar, OH spectrometer, satellite or rocket climatology. Hocking et al. (2001b) provided a *statistical comparison technique (SCT)* to calibrate the biased slope estimate as,

$$\beta_{OLS}^d = \left(1 - \frac{s_\delta}{s_d}\right) \beta_{SCT} \tag{7}$$





Or,

$$\beta_{SCT} = (1 - \frac{s_\varepsilon}{s_h}) \beta_{OLS}^h \tag{8}$$

where $s_\delta$ and $s_\varepsilon$ are the error variances of the (log of) diffusion coefficient (or decay time) and height respectively, $s_d$ and $s_h$ are data variances of $log_{10}(1/\tau)$ and of height respectively, $\beta_{OLS}^d$ and $\beta_{OLS}^h$ are slope coefficients when diffusion coefficient or
height is treated as independent variable respectively in the OLS regression analysis. The calibrated slope, $\beta_{SCT}$, is traditionally obtained by arbitrarily choosing $s_\delta$ or $s_\varepsilon$ that gives the best temperature estimates of MR as compared to optical or satellite data (e.g., Holdsworth et al., 2006; Hocking et al., 2007; Kim et al., 2012). A severe shortcoming of such calibration procedure is that the calibration factors ($s_\delta$ and $s_\varepsilon$) in the parenthesis of Eq. (7) or Eq. (8) are dependent on the data selection criteria (e.g., limiting heights, decay time or zenith angle to a certain range). Moreover, the outcome of any such calibration routine
will depend on the location of the MR and the choice of the calibration instrument. From a pure statistical context, the arbitrary choice of calibration parameters makes the estimated temperature also an arbitrary quantity thereby making it impossible to draw any reasonable statistical inferences.

In practice, the ordinary least-squares method will not be valid for MR data since neither the height nor the decay time can be predetermined as independent variable, and both variables are subjected to intrinsic measurement errors and various
geophysical effects. The reasons for such bias, and thus needing calibration, is discussed on theoretical and experimental grounds in Sect. 3.1. In addition, a statistical procedure to estimate $s_\delta$ and $s_\varepsilon$ using SCT calibration is formulated and presented in Sect. 3.1. An alternative method that includes measurement errors in the regression model is introduced in Sect. 3.2.1. This analysis does not require an absolute knowledge of $s_\delta$ or $s_\varepsilon$, but only the relative ratio is needed. As an alternative to SCT calibration, in Sect. 3.2.2, we have provided an analytic solution for estimating the bias-corrected slope coefficient. Using this
solution and Eq. (6), an independent and absolute value of MR temperature is obtained. A comparison study of the estimated MR temperatures with colocated lidar temperatures is discussed in Sect. 4.

## 2   Instrumentation and data

The All-Sky Interferometric Meteor Radar (SKiYMET) at Sodankylä Geophysical Observatory (SGO, 67°22' N, 26°38' E, Finland) has been routinely monitoring daily meteor-height averaged temperatures and wind velocity since December 2008
(Kozlovsky et al., 2016). The radar operates at a transmission power of 15 kW and frequency of 36.9 MHz, with a transmitting antenna which has a broad radiation pattern designed to illuminate a large expanse of the sky. The meteor trails are detected within a circle of 300 km diameter around SGO. The phase differences in the five-antennae receiving array allow the determination of the azimuth, elevation, range, and line-of-sight Doppler velocity of the meteor trails. The 2144 Hz pulse repetition frequency of MR transmission introduces a range ambiguity of 70 km, and the built-in analysis software therefore assumes
meteor trails are within the height of 70 to 110 km for unambiguous detections. Uncertainty in the height is ±1 km (or better for large zenith angle), which is determined by the 2 km range resolution. In addition, the half-time ($\tau$) of the received signal is calculated from the width of the autocorrelation function. A detailed description of the algorithm of the SKiYMET signal processing software is outlined in Hocking et al. (2001a).





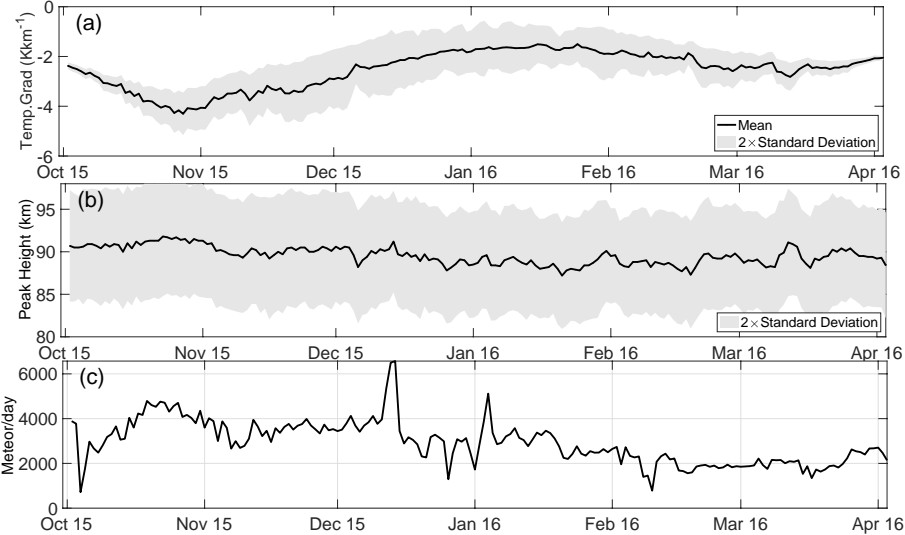

**Figure 2.** (a) Temperature gradient model derived from MSIS90. (b) Peak meteor heights for the data used in this work, and (c) the daily meteor detection for zenith angle less than $60°$ and velocity in the range $\pm100\,\mathrm{ms}^{-1}$.

SGO is located at the corrected geomagnetic latitude of $64.1°$, which is statistically a region of the auroral oval. Hence
the radar frequently detects nonmeteor targets during substorms associated with ionospheric plasma waves generated due to Farley-Buneman instability (Kelley, 2009). The Doppler velocity of such echoes can be more than a few $100\,\mathrm{ms}^{-1}$, which are mostly detected at low elevation (Lukianova et al., 2018). MR also detects ground echoes modulated by the ionosphere during pulsating auroras (Kozlovsky and Lester, 2015). These targets have near-zero Doppler velocities, and are also observed at low elevation. Our data selection criteria is kept to bare minimum, such that, all heights and decay times are included as long as
they are unambiguous detections above $30°$ elevation angle and Doppler radial velocity in the range $\pm100\,\mathrm{ms}^{-1}$.

For temperature estimation we considered one day of data for a six month period from October 2015 to March 2016 since simultaneous lidar measurements were available during this time. The Compact Rayleigh Autonomous Lidar (CORAL) provided vertical profiles of the atmospheric temperature at 27–98 km over Sodankylä as part of the GW-LCYCLE-II (Gravity Wave Life cycle Experiment) campaign in winter 2015/2016 (Reichert et al., 2019). The median number of daily meteor detections
in this data set is 2857, with a minimum of 716 meteor detections on 04 Oct 2015 and a maximum of 6572 detections during the Geminids meteor shower on 14 Dec 2015.

The Mass-Spectrometer-Incoherent-Scatter or MSIS90 (Hedin, 1991) model temperatures are used to generate a temperature gradient model near the peak heights. Model temperatures are computed for each date at intervals of 6 h between 85 km to 95 km. A third-degree polynomial fit is carried out to obtain the height profiles at 6 UT, 12 UT, 18 UT and 24 UT. For each time
interval, the gradient at the respective meteor peak height, as well as at 1 km above and below from peak height are estimated. These 12 values are then used to obtain the mean and standard deviation of the temperature gradient for each day near the peak height, which varies in the range $89\pm1$ km. The daily meteor detections, the peak heights, and the corresponding temperature



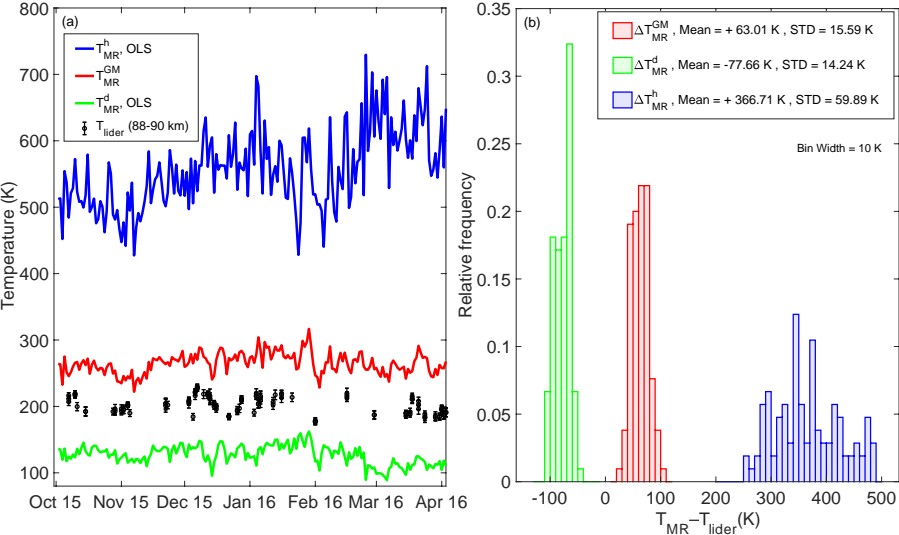

**Figure 3.** (a) Temperature offset in OLS method using $log_{10}(1/\tau)$ (*green*) and height (*blue*) as independent variable. Also, showing (*red*) the overestimated temperatures obtained using the geometric mean (GM) fitting. (b) The difference between the lidar ($T_{lidar}$) temperatures and the estimated MR temperatures ($T_{MR}$) using OLS fitting and GM fitting.

gradient model values are shown in Fig. 2. The standard deviation of the MSIS model values correspond to roughly of the order of $0.7~\mathrm{Kkm}^{-1}$.

Our reasons for choosing the MSIS temperature gradient are twofold. Firstly, MSIS data are easily accessible from the online version, which guarantees reproducibility of this work independent of location. Secondly, even if the temperature gradient term in Eq. (6) is ignored, the resulting offset in the estimated temperature can be no more than $10\%$ (Hocking, 1999). Hence an approximate estimate is sufficient for the main objective of this paper. However, the actual temperature gradient in the atmosphere may be slightly different from these model values, which can contribute to the biasing effect in the estimated

temperatures. Any such possibility and its effect on the estimated temperatures is addressed in the subsequent section.

## 3   Method: Regression analysis

### 3.1   Estimation of error variances in decay time and height

In the following text we use the notation and formulation in Gillard and Iles (2005). The observables, $log_{10}(1/\tau)$ and height, are represented as $d_i$ and $h_i$ respectively, and the corresponding unobserved true values as $\xi_i$ and $\eta_i$ respectively, where the

index $i$ represents the i'th meteor detection. For consistency we also assume that $log_{10}(1/\tau)$ is presented in abscissa and height is in the ordinate in the respective scattered plot (as shown in Fig. 1). Suppose, we are assuming a linear relation in variables $\xi_i$ and $\eta_i$ as





$$\eta_i = \alpha + \beta \xi_i, \ i = 1, 2, \ldots, N \tag{9}$$

Due to measurement errors and various geophysical processes, the true values $\xi_i$ and $\eta_i$ will be subjected to random errors and hence the observable $d_i$ and $h_i$ will have scatter around the linear model in Eq. (9):

$$d_i = \xi_i + \delta_i \tag{10}$$

And,

$$h_i = \eta_i + \varepsilon_i = \alpha + \beta \xi_i + \varepsilon_i \tag{11}$$

where $\delta_i$ and $\varepsilon_i$ are errors in the measured $log_{10}(1/\tau)$ and height respectively, and are assumed to be mutually uncorrelated, have zero mean and independent of the suffix $i$. This implies that the measurement error variances, $s_\delta$ and $s_\varepsilon$, are constant with respect to the suffix $i$. Classical regression analysis or ordinary least-squares (OLS) treats $d_i$ as an independent variable without intrinsic errors (or, $d_i = \xi_i$), and then minimises the sum of squared residuals along the ordinate. The slope from the OLS method can be represented in terms of the covariance ($Cov$) and variance ($Var$) as (e.g., Smith, 2009; Keles, 2018),

$$\beta^d_{OLS} = \frac{Cov(d_i, h_i)}{Var(d_i)} \equiv r \frac{\sqrt{s_h}}{\sqrt{s_d}} \tag{12}$$

where $\beta^d_{OLS}$ is the OLS slope estimated by considering $log_{10}(1/\tau)$ as independent variable, and $r$ is the Pearson product-moment correlation coefficient between $d$ and $h$. Likewise, by reversing the arguments above, it's trivial to show that the reciprocal value of OLS slope estimate with height as independent variable is (e.g., Smith, 2009),

$$\beta^h_{OLS} = \frac{Var(h_i)}{Cov(d_i, h_i)} \equiv \frac{1}{r} \frac{\sqrt{s_h}}{\sqrt{s_d}} \tag{13}$$

To see the effect of errors in the independent variable on the OLS slope estimate, e.g., for $\beta^d_{OLS}$, Eq. (10) and Eq. (11) are used in Eq. (12),

$$\beta^d_{OLS} = \frac{Cov(\xi_i + \delta_i, \alpha + \beta \xi_i + \varepsilon_i)}{Var(\xi_i + \delta_i)} \tag{14}$$

Since $\delta_i$, $\varepsilon_i$ and $\xi_i$ are mutually independent, Eq. (14) simplifies to ,

$$\beta^d_{OLS} = \frac{\beta \, Cov(\xi_i, \xi_i)}{Var(\xi_i + \delta_i)} = \frac{1}{1 + \frac{Var(\delta_i)}{Var(\xi_i)}} \beta = \zeta \, \beta \tag{15}$$

where $\zeta$ is known as the *attenuation* or *regression dilution bias*. Since variances are always positive by definition, Eq. (15) shows that in the presence of measurement error in the so-called independent variable (in abscissa), the OLS slope estimate ($\beta^d_{OLS}$) will always be smaller than the unbiased slope $\beta$. Likewise, $\beta^h_{OLS}$ is greater than $\beta$ if there is error in the measured height (specific example presented in Fig. 1a). By substituting $d_i = \xi_i + \delta_i$, we can rearrange Eq. (15) as

$$\beta^d_{OLS} = (1 - \frac{Var(\delta_i)}{Var(d_i)}) \beta \tag{16}$$





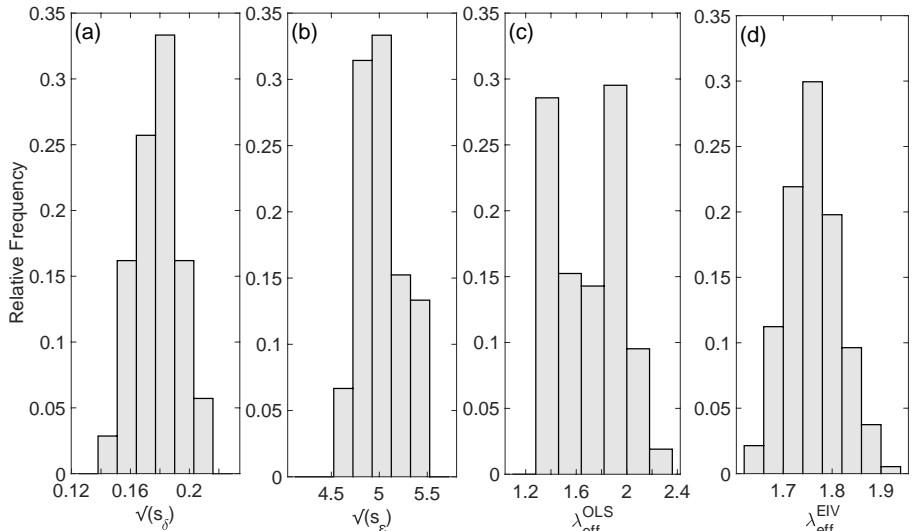

**Figure 4.** Statistical distribution of (a) $\triangle log_{10}(1/\tau)$ and (b) $\triangle height$ obtained using SCT method with colocated lidar data. The variance ratios of these parameters in the normalised coordinate system is shown in (c). For comparison, (d) presents these variance ratios obtained using EIV method (Sect. 4) for the same data set. The properties of these histograms are presented in Table 1.

Equation (16) is a well known identity (e.g., Carroll and Ruppert, 1996; Frost and Thompson, 2000) in statistical literature, that was re-derived by Hocking et al. (2001b) in Eq. (7) in the context of SCT correction. Equation (16) reveals that an absolute knowledge of error variances, $s_\delta$ ( or $s_\varepsilon$), is required to obtain the bias-corrected slope ($\beta$) if we choose OLS fitting for the slope estimate. To this date, no such attempt has been made to assess these error variances in MR data. Instead, biased temperatures were calibrated with optical or satellite data by arbitrarily choosing a value of $s_\delta$ or $s_\varepsilon$ for estimating the so-called SCT

slope, $\beta_{SCT}$ (e.g., Holdsworth et al., 2006; Hocking et al., 2007; Kim et al., 2012). In the remaining part of this section, we demonstrate how to obtain a representative value of the error variances in this data following a revised calibration procedure.

      For each 24 h of the data set, we performed two OLS fittings to estimate $\beta^d_{OLS}$ and $\beta^h_{OLS}$ by using Eq. (12) and Eq. (13). The corresponding biased temperatures, $T^d_{MR}$ and $T^h_{MR}$ respectively, are estimated using Eq. (6). Experimental values of the parameters $s_\delta$ and $s_\varepsilon$ can be obtained by comparing these estimated biased temperatures with the colocated lidar temperatures

($T_{lidar}$) as the reference values. Using Eq. (7) and Eq. (8), and noting that the slope is proportional to the estimated temperatures from Eq. (6) we obtain:

$$s_\delta \approx (\frac{T_{lidar} - T^d_{MR}}{T_{lidar}}) s_d \text{ and } s_\varepsilon \approx (\frac{T^h_{MR} - T_{lidar}}{T^h_{MR}}) s_h \tag{17}$$

where $T^d_{MR}$ and $T^h_{MR}$ are MR temperatures estimated using OLS fitting with $log_{10}(1/\tau)$ and height as independent variable respectively. Furthermore, if the measurements are normalised with the mean and standard deviation (STD) as

$$d'_i = \frac{d_i - mean(d_i)}{\sqrt{s_d}} \text{ and } h'_i = \frac{h_i - mean(h_i)}{\sqrt{s_h}} \tag{18}$$





then, $Var(d'_i) = Var(h'_i) = 1$, and the OLS estimate of the ratio of the measurement error variance is (from Eq. (17) and Eq. (18)):

$$\lambda_{eff}^{OLS} = \frac{s_{\varepsilon'}}{s_{\delta'}} \approx \frac{T_{MR}^h - T_{lidar}}{T_{lidar} - T_{MR}^d} \cdot \frac{T_{lidar}}{T_{MR}^h} \tag{19}$$

where the error variances, $s_{\varepsilon'}$ and $s_{\delta'}$, are in the dimension free system defined by Eq. (18). In essence, $\lambda_{eff}^{OLS}$ is a measure of
all sources of errors in the normalised heights and decay times that cause the real data to deviate from the idealized physical model of Eq. (6), thereby producing a typical scatter as seen in Fig. 1.

$\lambda_{eff}^{OLS}$, $s_\delta$, and $s_\varepsilon$ were estimated by Eq. (17) and Eq. (19) using 24 h of MR data and colocated lidar temperatures at 88 km, 89 km and 90 km for the dates for which lidar data was available. The biasing effect on the OLS estimate of MR temperatures with $log_{10}(1/\tau)$ and height as independent variable respectively are presented in Fig. 3 (green and blue lines and histogram).
As expected from Eq. (7) or Eq. (8), a mean offset of -78 K and +367 K occurs depending on whether $log_{10}(1/\tau)$ or height respectively is considered as independent variable. In practice, the magnitude of these biases is related to the total errors in heights and $log_{10}(1/\tau)$ from Eq. (17), which was not taken into account by the OLS regression model in Eq. (12) and Eq. (13). The experimental values of the error in height ($\sqrt{s_\varepsilon}$) and the error in log of inverse decay times ($\sqrt{s_\delta}$) and their ratios, spread over one winter season (2015–2016), are shown in Fig. 4. The mean and standard deviation of the distributions are
given in Table 1. The obvious best choice of $\sqrt{s_\delta}$ and $\sqrt{s_\varepsilon}$ are 0.18 and 5.04 respectively (Table 1) for calibrating SGO's MR temperatures using OLS method against colocated lidar temperatures. As an example, the calibrated temperatures obtained with $\sqrt{s_\delta} = 0.18$ and $\beta_{OLS}^d$ is presented in Sect. 4.

It is worth noting that instead of directly using the individual observation between biased MR temperatures and lidar measurements from Eq. (17), we have used the statistical mean of differences for calibration. This is because lidar data is not
available for all days during the 6 months of data used in this work. Moreover, both MR and lidar data have their own intrinsic errors and technical differences in the observation time and volume of sky. MR temperatures are daily averages over 24 h of observation, whereas lidar data is just the nightly mean profile. The lidar probes a small volume limited to the diameter of the lidar beam, while the radar illuminates a large part of the sky. For a *single* observation, lidar may see the phase structure of large-scale gravity waves while the MR averages over the gravity wave structure due to different spatial resolutions. As a
result, the radar averages over gravity waves with horizontal wavelengths smaller than few hundred km. On the other hand, the lidar may resolve these gravity waves if the run time is shorter than the period of these waves. As gravity wave amplitudes are on the order of 10–15 K at these altitudes, we cannot expect perfect agreement between radar and lidar temperature due to geophysical variations which show up differently in the two data sets as a result of the different observational volumes.

While such calibration routine may prevent large offsets in the estimated temperatures, the day-to-day variation in these
error variances (as shown in Fig. 4) due to natural geophysical processes will persistently introduce artefacts in the estimated temperatures. Moreover, due to the continually changing atmospheric dynamic, these calibration parameters need to be updated at time intervals. This in turn requires availability of optical or satellite data throughout the year. In the following section, we provide a robust method of estimating MR temperatures that do not require any external calibration.





**Table 1.** Representative value of the (square root of) error variances and their normalised ratio for SGO's meteor radar obtained by SCT method (Fig. 4) for winter 2015–2016. The literature values are from [1]Hocking (2004), [2]Holdsworth et al. (2006), and [3]Kim et al. (2012). For comparison, the EIV estimate of $\lambda$ is included in the table (see Sect. 4).

| – | Mean | STD | Literature Values |
|---|---|---|---|
| $\triangle(log_{10}(1/\tau))/\text{s}^{-1}$ | 0.18 | 0.02 | $0.14^{1,2}$ |
| $\triangle(height)/\text{km}$ | 5.02 | 0.27 | $3.25^1, 1.1^2, 1.0^3$ |
| $\lambda_{eff}^{OLS}$ | 1.70 | 0.30 | – |
| $\lambda_{eff}^{EIV}$ | 1.76 | 0.06 | – |

## 3.2 Errors-in-Variables (EIV) model

For fitting a straight line model, such as,

$$y(x) = a + bx \tag{20}$$

to a set of $N$ data points $(x_i, y_i)$ measured with errors, the corresponding $\chi^2$ merit function is (Press et al., 1992, p 660)

$$\chi^2 = \sum_{i=1}^{N} \frac{(y_i - a - bx_i)^2}{\sigma_{yi}^2 + b^2 \sigma_{xi}^2} \tag{21}$$

where $\sigma_{yi}$ and $\sigma_{xi}$ are the standard deviation of the $i$'th data point, and the weighted sum in the denominator of Eq. (21) can
be interpreted as the weighted-error of the $i$'th data point. The regression coefficients, $a$ and $b$, can be found by minimising the merit function with respect to these coefficients following any suitable numerical root-finding routine. However, under certain assumptions on the error variances, it is possible to derive an analytic solution for the regression coefficients. This analytic solution is introduced and discussed in Sect. 3.2.1. And in Sect. 3.2.2, we have looked more closely into the nature of various error variances in the data to construct a closed-form solution of the bias-corrected slope coefficient.

### 3.2.1 First estimate of slope ($\beta$)

Application of Eq. (21) to physics data requires that all measured variables are dimensionally consistent so that $\chi^2$ is dimension free. Moreover, the analysis in this section requires that the measurements are presented in an appropriate dimension free system. This facilitate the direct comparison between different parameters, such as, the measured variables or the associated error variances. By applying the coordinate transformation introduced in Eq. (18) to Eq. (9), we therefore intend to solve the
simplified bivariate linear system of equations,

$$\eta_i^{'} = \beta \frac{\sqrt{s_d}}{\sqrt{s_h}} \xi_i^{'} \equiv \beta_W \xi_i^{'} \tag{22}$$

where $\eta_i^{'}$, $\xi_i^{'}$ and $\beta_W$ are dimension free. For the specific choice of normalisation by Eq. (18), the intercept ($\alpha_W$) is always zero in the transformed coordinate system. The merit function in Eq. (21) can be further simplified by invoking a *homoscedastic*





*standard weighting model* (Macdonald and Thompson, 1992). This error model assumes that the error variances are indepen-
dent of data point, thereby simplifying the merit function as (Macdonald and Thompson, 1992; Lolli and Gasperini, 2012),

$$\chi^2(\beta_W, \alpha_W = 0) \equiv \sum_{i=1}^{N} \frac{(h_i' - \beta_W d_i')^2}{s_{\delta'}(\lambda + \beta_W^2)} \tag{23}$$

where $s_{\delta'}$ and $s_{\varepsilon'}$ are *constant* error variances of the measured $log_{10}(1/\tau)$ and heights respectively in the normalised (or,
dimension free) coordinate system, and $\lambda$ is the ratio

$$\lambda = \frac{s_{\varepsilon'}}{s_{\delta'}} \tag{24}$$

The $\chi^2$ minimisation of Eq. (23) with respect to $\beta_W$ leads to the well-known (Carroll and Ruppert, 1996; Smith, 2009; Lolli
and Gasperini, 2012) analytic expression for the EIV slope parameter in terms of the variances ($s_{d'}$, $s_{h'}$) and covariances
($s_{h'd'}$) of the measured variables,

$$\beta_W = \frac{s_{h'} - \lambda s_{d'} + \sqrt{(s_{h'} - \lambda s_{d'})^2 + 4\lambda s_{h'd'}^2}}{2s_{h'd'}} \tag{25}$$

Or,

$$\beta_W \equiv \frac{1 - \lambda + \sqrt{(1 - \lambda)^2 + 4\lambda s_{h'd'}^2}}{2s_{h'd'}} \tag{26}$$

Since $s_{d'} = s_{h'} = 1$ from Eq. (18). And the covariance ($s_{h'd'}$) is computed using the standard definition,

$$s_{h'd'} = \frac{1}{N} \sum_{i=1}^{N} (h_i' d_i'), \text{ for large } N \tag{27}$$

Equation (26) can be solved if a priori knowledge of $\lambda$ is available, which in turn requires a precise estimate of all sources
of errors in the measured data. In the more practical case for unknown $\lambda$, we need to initiate a good starting estimate. Using
the calibration procedure described by Eq. (17) and Eq. (18), the mean values of $s_{\varepsilon'}$ and $s_{\delta'}$ are found to be $0.64 \pm 0.04$ and
$0.38 \pm 0.06$ respectively. Since the EIV estimate of the slope requires only the ratio between $s_{\varepsilon'}$ and $s_{\delta'}$, a good choice of
this starting value is $\lambda = 1$. Furthermore, for $\lambda = s_{\varepsilon'} = s_{\delta'} = 1$, there exists a simple geometric interpretation of the merit
function in Eq. (23). This solution corresponds to minimising the euclidean or orthogonal distance between the fitted line and
the measured data. The residual function to be minimised with respect to the regression coefficients is

$$\chi^2 = \sum_{i=1}^{N} \frac{(h_i' - \beta_W d_i')^2}{1 + \beta_W^2} \equiv \sum_{i=1}^{N} \frac{(h_i' - d_i')^2}{2} \tag{28}$$

since $\beta_W = 1$ when $\lambda = 1$ from Eq. (26). Following Eq. (22), we therefore have our *first estimate* of $\beta$ in the scattered plot of
$log_{10}(1/\tau)$ and heights,

$$\beta = \frac{\sqrt{s_h}}{\sqrt{s_d}} \tag{29}$$



Equation (29) is commonly referred to as reduced major axis (RMA) solution in statistics literature (Smith, 2009). In practice, this is just the geometric mean (GM) of the two OLS estimates, $\beta_{OLS}^d$ and $\beta_{OLS}^h$, as can be seen by combining Eq. (12) and Eq. (13),

$$\beta_{GM} = \sqrt{\beta_{OLS}^d \, \beta_{OLS}^h} \equiv \frac{\sqrt{s_h}}{\sqrt{s_d}} \tag{30}$$

The GM solution in Eq. (30) has the unique feature that this is the only case of EIV estimate which is both scale-invariant
and symmetric in the variables (e.g., Ricker, 1984; Smith, 2009). While these properties do not necessarily imply that the GM solution is the correct solution (discussed below), but this *first estimate* is essential to solve the problem of biasing in the estimated slope (Sect. 3.2.2). A specific example of GM fitting is presented in Fig. 1a.

The systematic offset between MR temperature and colocated lidar temperature as a result of using Eq. (30) is shown in Fig. 3a (red curve) and Fig. 3b (red histogram). The MR temperature is estimated with 1 day of radar data. Each of these
temperatures are then compared with lidar temperatures at 88 km, 89 km and 90 km for the dates when lidar data are available. The intrinsic noise in the lidar temperature is about 5–10 K (Reichert et al., 2019), which implies no temperature gradient is observed between 88–91 km in these data. Figure 3b (red histogram) reveals that the MR temperatures are over-estimated by a mean value of +63 K for the case of GM solution .

When compared to the temperature-gradient model derived from optical, satellite and rocket climatology (e.g., Holdsworth
et al., 2006), it can be easily argued that our MSIS derived gradient model values (Fig. 2a) is more negative than expected. If these values are shifted by a constant positive offset of +1 Kkm$^{-1}$, the absolute value of the estimated temperatures will increase by 10 K (Singer et al., 2004). This will further increase the offset between lidar and MR temperature, thereby shifting the histogram (in red) in Fig. 3b further to the right.

On the other hand, lidar temperatures are usually obtained during the night time, which can lead to a systematic offset
due to day-night differences or tidal variations. As discussed by Hocking et al. (2004), the day-night temperature difference at these altitudes is of the order of 3–4 K. This is significantly less than the standard errors in these temperatures, which is on average 19 K. Hence we expect the day-night difference in SGO's MR temperatures to be insignificant during the winter period. Moreover, any attempt to estimate MR temperatures using only night-time data has the adverse effect of reducing the accuracy in the estimated temperatures due to data loss. While no specific studies of tidal variation have been made for this
location, the data from other sites (e.g., Hocking and Hocking, 2002; Stober et al., 2008) show that the temperature variation due to tidal activity is typically less than 10 K. We can therefore rule out the possibility of an offset in the MSIS gradient model or tidal effects as the primary cause for the +63 K offset seen in Fig. 3 (red curve and histogram).

Ricker (1984) emphasised that the biasing in GM solution is conditional upon the value of the correlation coefficient ($r$) between the variables, while Kimura (1992) demonstrated that this solution will be an overestimate in the case of low $r$ value.
For the data set used in this work, we found that the correlation between $log_{10}(1/\tau)$ and height is typically $0.50 \pm 0.05$, thereby indicating the presence of significant natural variation in the measurements. Furthermore, Jolicoeur (1990) used error modelling to conclude that $r$ must be more than 0.6 for the the GM solution to be acceptable. In fact, we have observed that if we restrict our data selection process by excluding all data beyond the density contour 0.2 (Fig. 1b), this artificially increases





the $r$ to be typically around $0.66 \pm 0.06$ with the consequence of reduced biasing in the GM solution. However, temperatures
estimated with any such arbitrary choice of data rejection criteria will lack consistency.

From purely mathematical perspective, Eq. (25) provides two equivalent explanations for the overestimate in GM solution. Either the assumption of $\lambda = 1$ is not correct, and thus a higher value of $\lambda$ is required to correct for the bias. This was validated in Sect. 3.1 using SCT calibration, where we have obtained $\lambda_{eff}^{OLS} = 1.70 \pm 0.30$ (Table 1). Or, the assumption of $\lambda = 1$ is correct, but the effective variance of the normalised heights, $s_{h'}$, is less than 1. This can be achieved by introducing a *correction term*
in the statistical model of Eq. (22). In Sect. 3.2.2, we follow the second approach. And in Sect. 4, we have combined both approaches to obtain a numerical estimate of the effective value of $\lambda$.

### 3.2.2   Reweighted second estimate of slope ($\beta$)

Equation (26) indicates that the overestimate of the GM solution can be corrected by using an increased value of $\lambda$, although it's an unknown parameter. Carroll and Ruppert (1996) attributed the parameter $\lambda$ as purely due to measurement errors alone.
Following such argument, Carroll and Ruppert (1996) criticised Eq. (22) as an incomplete model for real data, since it implies that in absence of measurement error only, *"the data would fall exactly on a straight line"*. In practice, meteor echoes are subjected to random variations due to natural geophysical processes at all heights. Hence an additional term, known as *equation error* $(\mu')$, is necessary to account for such geophysical or natural variation in the data. Therefore, a correction term $(\mu_i')$ is introduced in the statistical model (Carroll and Ruppert, 1996, Sect. 3),

$$\eta_i' = \beta_W^{adj} \, \xi_i' + \mu_i' \tag{31}$$

The adjusted value of the slope $(\beta_W^{adj})$ in presence of equation error is always less than $\beta_W$ in the statistical model of Eq. (22). This can be seen by expressing Eq. (22) and Eq. (31) in terms of the variances,

$$\sqrt{\frac{s_{\eta'} - s_{\mu'}}{s_{\xi'}}} < \sqrt{\frac{s_{\eta'}}{s_{\xi'}}}, \text{ Or } \beta_W^{adj} < \beta_W \tag{32}$$

For the specific case of GM fitting or for $\lambda = 1$, Eq. (32) implies $\beta_W^{adj} < 1$. This solution requires that the observed variance of
the measured heights in the data need to be weighted down by an appropriate scaling factor to compensate for the presence of natural variation in the data. Alternatively, the effective value of $\lambda$ in presence of equation error can be written as (Carroll and Ruppert, 1996),

$$\lambda_{eff}^{EIV} = \frac{s_{\varepsilon'} + s_{\mu'}}{s_{\delta'}} \tag{33}$$

The quantity $s_{\mu'}$ essentially incorporates additional variability in the data that doesn't conform to the original construct or
the characteristics being measured according to the underlying assumptions of Eq. (6). For example, meteor trails can get modified by wind effects, ion composition, meteor fragmentation, strong ionospheric currents as well as temperature and pressure fluctuations on various spatial and temporal scale (Hocking, 2004; Younger et al., 2014). Hence the practical effect of the extra term in Eq. (31) is to increase the effective variance of $h_i'$ by an additional factor of $s_{\mu'}$ (Gillard and Iles, 2005).





Assessing the sampling distribution of $s_{\mu'}$ would require carefully designed replicates of observations as well as long term
comparison of MR temperatures with other colocated instruments. This line of reasoning limits the practical use of Eq. (33).
Because an absolute knowledge of measurement error or various sources of natural variation is not available for this data on a
day-to-day basis.

Smith (2009) provided an alternative, but equally valid interpretation of Eq. (33) in connection to the biasing effect on the
GM solution. According to Smith (2009), Eq. (33) is merely a convenient choice of error-partitioning. It is not the absolute
value of equation error variance but rather the relative magnitude of this error that determines the biasing effect. Moreover,
Smith (2009, p 481) argues:

" Natural variation is not a characteristic of any single measured value. It exists only in the context of full analytical model,
and that model is defined by the investigator."

In contrast to Carroll and Ruppert (1996), we therefore no longer stringently distinguish between different types of errors
in the data (e.g., Sprent, 1990, p 13). In other words, the error variances, $s_{\delta'}$ and $s_{\varepsilon'}$, now consist of both measurement errors
and partly the natural geophysical variation. Hence the equation error variance, $s_{\mu'}$ in Eq. (33), is simply a representative of
the asymmetric component of the natural variation in the data. In MR data, such asymmetric effect of natural variation occurs
mainly at altitudes approximately above 95 km due to geomagnetic effect, and approximately below 85 km due to electron
recombination and attachment (Sect. 1 and the references therein). To compensate for this asymmetric component of natural
variation in the data, the observed variance of the normalised heights ($h_i'$) must be replaced by its reweighted value ($h_i^*$) as,

$$Var(h_i^*) = Var(\nu h_i') = \nu^2 \tag{34}$$

such that the scaling factor, $\nu$, is less than 1. In the Appendix section we have derived an analytic solution of $\nu$ in terms of $s_{\mu'}$:

$$\nu^2 = 1 - s_{\mu'} = 1 - Var\left(\sqrt{\frac{(h_i' - d_i')^2}{2}}\right) \tag{35}$$

For $\lambda = 1$, Eq. (25) can be expressed in terms of the reweighted normalised heights as ,

$$\beta_W^{adj} = \frac{s_{h^*} - s_{d'} + \sqrt{(s_{h^*} - s_{d'})^2 + 4s_{h^*d'}^2}}{2s_{h^*d'}} \tag{36}$$

Using Eq. (34) in Eq. (36), the bias-corrected slope is,

$$\beta_W^{adj} = \frac{\nu^2 - 1 + \sqrt{(\nu^2 - 1)^2 + 4\nu^2 s_{h'd'}^2}}{2\nu s_{h'd'}} \tag{37}$$

where $\nu$ can be estimated from Eq. (35). Equation (37) is therefore the reweighted *second estimate* of the normalised slope.
It is to be noted that $\lambda$ has been essentially eliminated in Eq. (37), and hence an absolute knowledge of the ratio of error
variances is no longer necessary as a result of data normalisation. Equation (35) and Eq. (37) can be directly used to obtain the
bias-corrected slope using the measured values of decay times and heights.

The line-fitting procedure developed in this section can be summarised as follows: First, the values of $log_{10}(1/\tau)$ and heights
for each 24 h of data are normalised using Eq. (18). Second, Eq. (27) and Eq. (35) are used for computing $s_{h'd'}$ and $\nu$. Third,





the values of $s_{h'd'}$ and $\nu$ are substituted in Eq. (37) to obtain the normalised slope coefficient ($\beta_W^{adj}$). In the final step, the slope

and the intercept in the original coordinate system is calculated from,

$$\beta = \beta_W^{adj}(\sqrt{s_h}/\sqrt{s_d}) \tag{38}$$

And,

$$\alpha = \bar{h} - \beta \times \bar{d} \tag{39}$$

This value of $\beta$ and the temperature gradient (Fig. 2) are used in Eq. (6) to obtain the bias-corrected MR temperature of the

atmosphere at the height of peak meteor occurrences. Examples of slope estimates using various regression models discussed

in Sect. 3.1 and Sect. 3.2 are presented in Fig. 1a for comparison. And in Sect. 4, we have provided both experimental and

numerical validation of the use of Eq. (37) for estimating the MR temperatures.

We conclude this section with the following remarks: The EIV solution requires *a priori* knowledge of $\lambda$. But the question

regarding what exactly is this value, can be answered only in the context of what statistical model we are considering. This is

the key theoretical aspect from which we have formulated the analytic solution of the slope coefficient in Eq. (37).

### 3.2.3    Bootstrap analysis of the standard error in $\beta$

Monte Carlo simulation is carried out to assess the standard error in the estimated $\beta$ (Press et al., 1992, Sect. 15.6). The actual

daily data set, with its $N$ data points, are used to generate 20000 synthetic data sets using random resampling. Each of these

synthetic data set is made by randomly drawing $N$ data points with replacement from the original data set, while allowing

repeated draw of the same data. Hence for large $N$, approximately $e^{-1}th$ fraction of the actual measurements are replaced by

*duplicated* original points. As mentioned above we have repeated the resampling procedure 20000 times and then assessed

the standard error (SE) of the parameters, $Var(\beta_W^{adj} h_i)$ and $Var(d_i)$, from the 95% confidence interval (CI) of the distribution

as,

$$SE = \frac{CI}{2 \times 1.96} \tag{40}$$

Equation (38) can be used to compute the standard error in $\beta$ from the standard error in $\beta_W^{adj}\sqrt{s_h}$, $\sqrt{s_d}$ and the algebra of error

propagation. Figure 5 presents the error estimates in $\beta$ for the data sets used in this paper. As expected, the estimated standard

error was found to be inversely related to $\sqrt{N}$. A linear fit revealed the following relation,

$$\Delta\beta \times \sqrt{N} = 299\% \tag{41}$$

where $\Delta\beta$ represents the standard error as a percentage of the estimated $\beta$. According to Eq. (41), for $N$ equal to 2000, we

expect the precision in the estimated slope to be about 7%. However, this value is quite conservative. For most days of the year,

the daily number of meteor detections usually exceeds 2000 by far, hence we expect the percentage errors in the estimated $\beta$

to be typically less than 7%.





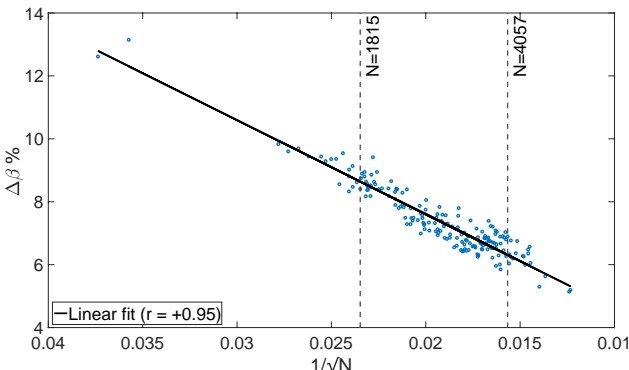

**Figure 5.** Estimated percentage errors in the slope using bootstrap resampling. The two dashed lines indicate the range of $N$ corresponding to $80\%$ or 148 days out of the total of 185 days of data used in this work.

## 4   Results and Discussion

We have applied the method formulated in Sect. 3.2.2 to the MR data from the period October 2015 to March 2016. This is

presented in Fig. 6, along with the data from colocated lidar observation. The percentage differences between MR temperatures and lidar are shown in Fig. 7a for altitudes near peak meteor counts. The mean difference between lidar and MR temperatures are less than $\pm 1$ K (or about $0.2\%$) which is negligible within error consideration. This means we have achieved obtaining an unbiased temperature estimation. The root-mean-square differences (RMS) is about $7\%$. This is expected since a $0.7$ $\mathrm{Kkm}^{-1}$ error in MSIS gradient model will introduce a $7$–$10$ K variation in the data for a typical winter temperature of $200$ K in the peak

meteor region. (Hocking et al., 2004; Meek et al., 2013). In addition, lidar temperatures have intrinsic errors of $5$–$10$ K. The $10\,percentile$ and the $90\,percentile$ value of daily number of meteor detections in this data set is 1815 and 4057 respectively. Using Eq. (41), this corresponds to an error in the estimated slope to be in the range of $5$–$7\%$. Hence the RMS difference of $7\%$ between MR temperature and lidar is not statistically significant within the error limits. In other words, we have managed to estimate MR temperatures correctly without requiring any external calibration.

The standard error in temperatures estimated using EIV analysis is on average 15 K. As can be seen in Fig. 6, around middle of Feb 2016 there is an abrupt change in temperature that is much larger than 2-sigma error bars. This temperature drop corresponds to a minor sudden stratospheric warming (SSW). Then, it was a "major final warming" starting on 5–6 Mar 2016 (e.g., Manney and Lawrence, 2016), which also coincides with a temperature decrease seen in this data. This demonstrates that there are day-to-day geophysical processes that can be revealed by the method developed in this work.

For direct comparison with the results above, we have also estimated the MR temperatures using the revised SCT calibration procedure described in Sect. 3.1. For this we have estimated the OLS slope ($\beta_{OLS}^d$) and used $\sqrt{s_\delta} = 0.18$ (Table 1) to obtain the calibrated temperatures from Eq. (16) and Eq. (6). These calibrated temperatures are presented in Fig. 6. For the reasons explained at the end of Sect. 3.1, the artifacts in the SCT calibrated temperatures are clearly visible in Fig. 6. The histogram in Fig. 7b shows the percentage differences between lidar data and the SCT calibrated MR temperatures. The mean difference





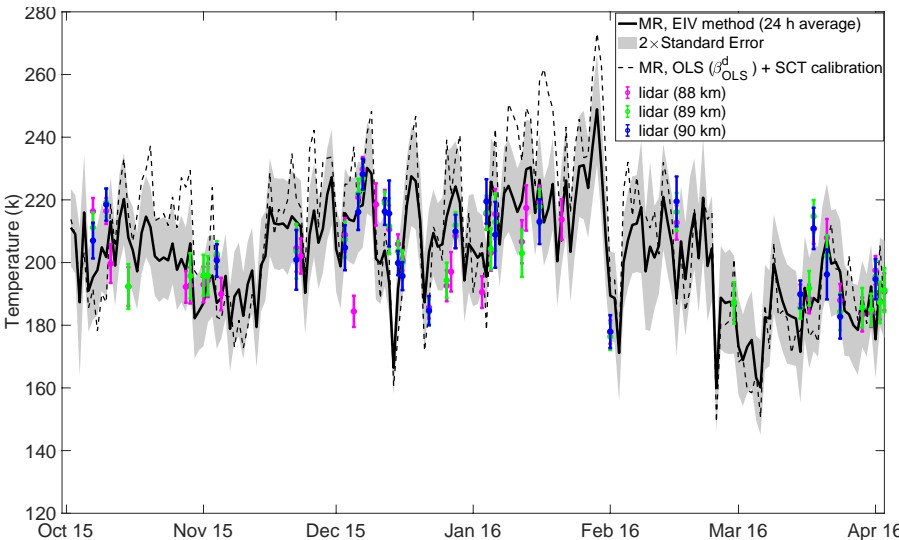

**Figure 6.** Comparison of the bias-corrected MR temperatures with lidar data for the winter 2015–2016. The *solid* line corresponds to the temperature estimated using the EIV method described in Sect. 3.2.2. The *dashed* line corresponds to the SCT calibrated temperatures using the colocated lidar measurements. The OLS estimates are obtained with $log_{10}(1/\tau)$ as independent variable. The errors in lidar temperatures are 5–10 K and the standard error (grey shade) of the temperature from EIV analysis is on average 15 K. The differences between lidar and MR temperatures are presented in Fig. 7.

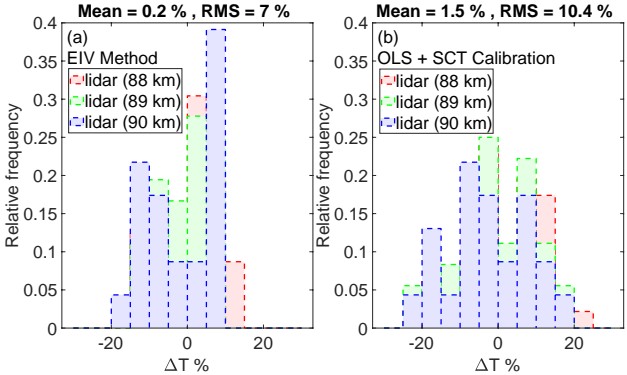

**Figure 7.** Percentage difference between MR temperatures and lidar data using (a) EIV method of Sect. 3.2.2 and (b) SCT calibration (Sect. 3.1) applied to OLS estimate of $\beta_{OLS}^d$. MR temperatures are shown in Fig. 6.

between the MR temperatures and that of lidar is again less than 2%, thereby showing that the biasing effect has been properly corrected by this calibration procedure. However, the RMS difference has increased to 10%, thereby clearly indicating the presence of artefacts in these calibrated MR temperatures.

As emphasised in Sect. 3.2, the statistical models in Eq. (22) and Eq. (31) are simply two equivalent representations of this data. Solution of the slope parameter for the model in Eq. (22) is given by Eq. (26). For *a priori* value of $\lambda = 1$, the




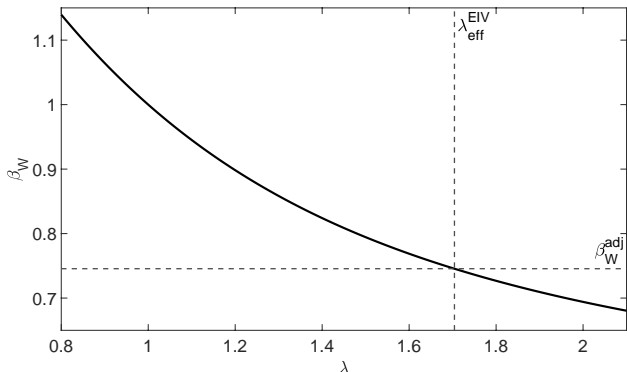

**Figure 8.** Two independent estimates of slope using EIV method for the date 14 Nov 2015. The line corresponds to the slope estimated using Eq. (26) with $\lambda$ as free parameter. $\beta_W^{adj}$ is the slope estimated from Eq. (37). Comparison of these two methods shows that the value of $\beta_W^{adj}$ is equivalent to using $\lambda = 1.7$ in Eq. (26).

slope estimated showed overcorrection and thus implying a higher value of $\lambda$ is required. Or equivalently, we have continued to assume $\lambda = 1$, and instead changed the statistical model to Eq. (31). This model effectively reduced the variance of the measured heights to allow a bias-corrected GM fitting. This solution is given by Eq. (37), which has been validated by the excellent agreement between lidar and MR temperatures (Fig. 6). Alternatively, we can show numerically that Eq. (37) is equivalent to Eq. (26) as follows:

First, the parameter $\lambda$ is allowed to run as free variable from 0.5 to 2.5 in steps of 0.01. Each value of $\lambda$ is then used in Eq. (26) to obtain a value of $\beta_W$. Second, the same data set is now used to obtain the bias-corrected slope using Eq. (37) for which we get $\beta_W^{adj} = 0.75$. The slope estimates from these two independent procedures are shown in Fig. 8 for one specific date. The value of $\beta_W^{adj} = 0.75$ corresponds to $\lambda_{eff}^{EIV} = 1.7$ when $\lambda$ is introduced as free variable in Eq. (26). Hence, the solution given by Eq. (37) for $\lambda = 1$ is essentially equivalent to increasing the effective value of $\lambda$ in Eq. (26).

The effective value of $\lambda$ (or, $\lambda_{eff}^{EIV}$ ) for each of the 24 h data set can estimated numerically by equating Eq. (26) with Eq. (37),

$$\beta_W^{adj} = \frac{1 - \lambda + \sqrt{(1-\lambda)^2 + 4\lambda s_{h'd'}^2}}{2s_{h'd'}} \tag{42}$$

The distribution of $\lambda_{eff}^{EIV}$ obtained from the full data set shows a mean and standard deviation of 1.76 and 0.06 respectively (Fig. 4d and Table 1). This is well within the limit of the experimental value of $\lambda_{eff}^{OLS}$=1.7±0.3 which we have obtained using

the SCT method (Fig. 4c and Table 1). Another implication of Eq. (37) can be seen in Fig. 3 (red line and the red histogram). The temperature estimated using the GM fitting (the *first estimate*) shows a systematic positive offset of +63±16 K. For a typical winter temperature of 200 K in the meteor regions (Hocking et al., 2004; Hall et al., 2006), this would correspond to a positive biasing of 23–40%. In the example provided for the sample data (Fig. 8), we see that use of Eq. (37) reduced the GM slope from 1 to 0.75 or by 25%. Thus, application of Eq. (37) has effectively reduced the offset by shifting the *red* histogram in

Fig. 3b leftwards without requiring any external calibration. This is presented in Fig. 7a where the mean offset is nearly zero.





Despite the excellent agreement in the statistical measure of ratios of error variances using two independent methods, $\lambda_{eff}^{OLS}$ and $\lambda_{eff}^{EIV}$, the estimated values of $\triangle log_{10}(1/\tau)$ and $\triangle(height)$ are clearly higher than what has been reported in literature previously (Table 1). Hocking (2004) estimated $\triangle$(height) = 3.25 km from pure measurement error perspective, using a numerical model for a pulse length equivalent to 2 km, and a meteor at an altitude of 90 km and zenith angle $50°$. Likewise, their

estimation of $\triangle log_{10}(1/\tau)$=0.14 was based on simulation studies for meteors below 95 km and confirmed that decay times variability arises due to 27% variability in $K_{amb}$ and 8% variability in temperatures over meteor region. The argument for choosing 95 km as the maximum height is that above this altitude meteor decay rates are substantially affected by processes other than ambipolar diffusion. Holdsworth et al. (2006) and Kim et al. (2012) also restricted their data selection process below the altitude of 95 km (or within 86–96 km in latter), and concluded that $\triangle$(height) $\approx 1.0$ km is the best value to calibrate their

MR temperatures with optical, satellite or rocket climatology using the SCT correction. In addition, Holdsworth et al. (2006, p 5) used various other data rejection criteria and assumed $\triangle log_{10}(1/\tau) = 0.14$ for calibrating the slope. In principle, these literature values of the error variances ( Table 1) were not derived following any statistical procedure, but rather assumed on the basis of speculation and convenience.

Our estimates of $\triangle log_{10}(1/\tau)$ and $\triangle(height)$ are higher, because we chose to include meteors at all altitudes and all

decay times in the data selection criteria. Due to geomagnetic effect the decay times are higher than expected from the simple ambipolar case at altitudes roughly above 95 km, whereas below 85 km the effect is the opposite (Sect. 1 and the references therein). More than 30% of the meteor echoes are from above 95 km and below 85 km. This means the value of $s_\delta$ and $s_\varepsilon$ will be higher due to increased geophysical variation in the data if no restriction is applied to the height range and decay times in the data selection process. By including meteor echoes from all altitudes we have avoided sampling bias, and hence we

believe our estimates of $s_\delta$ and $s_\varepsilon$ incorporate all sources of variation in the data, both instrumental and geophysical, thereby probing the true meteor population. Moreover, the altitude range where most meteor decays deviate from the assumption of ambipolar diffusion is a dynamical process, showing seasonal variation and can be location-dependent. In essence, there is no way to determine what are outlier in the data, since no error estimates are provided in the measured heights and decay times by the radar software. By keeping meteor echoes from all altitudes, we have retained most of the natural variation in the

measurements, and this ensures that the robustness of the proposed EIV method can be assessed to the fullest extent. Moreover, any arbitrary choice of data rejection criteria will also change the slope coefficient in an arbitrary way, and thus the estimated MR temperatures will lack consistency. To address this issue, in Sect. 3.2.2, we have incorporated the statistical effects of natural geophysical processes at all altitudes in the regression model.

Due to lack of colocated lidar measurements at other times of the year we have restricted our temperature estimation only

for the winter times. Complications might arise during certain days in the summer time when the turnaround of the diffusion coefficient profile occurs at higher altitudes close to the peak meteor regions. Hence the decay of meteor radar echos deviate significantly from diffusion-only evolution (Lee et al., 2013). This would require an improvement in the original physical model in Eq. (6), or some other alternative method of temperature estimation technique as discussed below.

In addition to the temperature-gradient technique used in this work, another common method for estimating MR temperatures

is to directly use some model pressures in Eq. (2). This method doesn't require any line-fitting analysis, and in theory, can be





used to determine temperatures at any heights within the meteor height distribution. The main drawback of this technique is that the pressure is a sensitive function of heights. Hence, even a small uncertainty in these model pressure values can lead to a large bias in the estimated temperatures thereby requiring further recalibration. Such biasing effect can be seen in Meek et al. (2013, p 1273). Using the pressure values from two different models, Holdsworth et al. (2006) recalibrated their temperatures as

$T_0 = 1.28T_{MR} - 52.60$ and $T_0 = 1.53T_{MR} - 109.7$ for the same data set. Likewise, Hall et al. (2006) used $T_0 = 1.12T_{MR} - 93$ for calibrating the MR temperatures against optical data. Later on, Dyrland et al. (2010) used the same data set as Hall et al. (2006), but calibrated their MR temperatures using satellite data and consequently derived a different calibration equation.

As an alternative to pressure and temperature-gradient methods, Hocking (2004) claimed that the correlation coefficient ($r$) between $log_{10}(1/\tau)$ and height is linearly related to the MR temperatures, and obtained $T_0 = 360r - 42$ for $T_0 < 190$ K. However, this method is based on the flawed assumption that a large value of correlation coefficient between $r$ and $T_0$ implies a linear relation between these two variables. Any such linear relationship between $r$ and $T_0$ must be validated with physics theory, which was not provided in Hocking (2004).

More recently, Lee et al. (2016) provided a good alternative technique for estimating MR temperatures near peak heights. Using both theory and experimental data, Lee et al. (2016) demonstrated a linear relation between the MR temperatures and the width of the meteor height distribution. Once the constant of proportionality of this linear correspondence is found, the MR temperatures can be directly estimated from the meteor height distribution. Lee et al. (2018) derived this proportionality constant empirically by using temperatures from satellite data, which in turn, is essentially a form of calibration process.

In all generality, it is desirable to avoid any kind of calibration process and instead, formulate an independent and unbiased estimate of temperatures using MR data alone. The statistical method presented in this paper pave the way towards achieving that goal.

## 5 Summary

The biasing effect in MR temperature has been a pressing issue for last two decades. Attempts have been made in the past to correct the slope in the scattered plot of $log_{10}(1/\tau)$ and height, usually either by direct calibration with optical or satellite data or by an arbitrary choice of data rejection criteria to exclude parts of measurements. This paper has addressed the underlying reasons for such biasing effect, which is mainly due to the presence of various error terms in the physical model. We have reviewed the conventional calibration procedure (1), and then provided an alternative but more robust method (2) for estimating MR temperature that doesn't require any calibration. We have applied both of these methods to the MR data from winter 2015–2016, and assessed the quality of the estimated MR temperature using colocated lidar measurements. The key points from each of these two aspects of the paper are given below.

1. This paper has reviewed the *statistical comparison technique* (SCT), originally proposed by Hocking et al. (2001b), within the context of MR temperature calibration. We have extended the theoretical basis of the SCT method to obtain an estimate of error variances of $log_{10}(1/\tau)$ and height using colocated lidar measurements. No significant offset was seen in the calibrated MR temperature, even without applying any outlier rejection criteria. But artefacts introduced due





495 to the difference in measurement techniques between MR and lidar was clearly visible in the estimated temperatures in the form of increased RMS differences.

2. As an alternative method, we have applied the *Errors-in-Variables* (EIV) regression model to estimate the slope in the scattered plot of $log_{10}(1/\tau)$ and height. This model takes into account the measurement errors in both abscissa and ordinate. Following Carroll and Ruppert (1996), an important source of variation, known as *equation error* was introduced
500 in the EIV method. The term *equation error* was shown to play a critical role in determining the MR temperature in the most unbiased way. The main cause of this error was identified as due to a number of geophysical effects in the measured data that was not taken into account in the original physical model. We have used the theoretical considerations by Carroll and Ruppert (1996) and Smith (2009) to formulate a closed-form analytic solution of the unbiased slope coefficient. This solution allows an independent estimate of atmospheric temperatures using meteor radar at the altitude of peak meteor occurrences. The temperatures estimated using this method show very good agreement with colocated lidar
505 measurement, with $7\%$ or better accuracy and without any systematic offset. The quality of the estimated temperatures using the EIV method was significantly better than the conventional SCT calibration method.

*Data availability.* The meteor radar data were collected at SGO (https://www.sgo.fi/Projects/SLICE/). Data used in the paper are available at https://www.sgo.fi/pub/AMT_2020_ESarkar/. The MSIS model data were obtained from https://ccmc.gsfc.nasa.gov/modelweb/. The lidar data are available at https://halo-db.pa.op.dlr.de/mission/109.

**Appendix A: Approximate analytic solution of $\nu$**

The revised statistical model of Eq. (31) implies that the variable $h_i^{'}$ is now redefined in its adjusted form $(h_i^{*})$,

$$h_i^{*} + \mu_i^{'} = \beta_W^{adj} \xi_i^{'} + \varepsilon_i^{'} = \beta_W^{adj}(d_i^{'} - \delta_i^{'}) + \varepsilon_i^{'} \tag{A1}$$

such that the expectation value of $\mu^{'}$, or $< \mu^{'} > \neq 0$, to account for the biasing effect due to asymmetric natural variation in the measured normalised heights. The variance of the reweighted heights are related to the original measurements as,

515 $$s_{h^{*}} = s_{h^{'}} - s_{\mu^{'}} = 1 - s_{\mu^{'}} \tag{A2}$$

By defining a parameter, $\nu$, such that,

$$\nu = \sqrt{1 - s_{\mu^{'}}} \tag{A3}$$

Equation (A2) can be expressed as,

$$Var(h_i^{*}) = Var(\nu h_i^{'}) = \nu^2 Var(h_i^{'}) = \nu^2 \tag{A4}$$



Under the assumption that the initial guess of $\lambda = 1$ is close to the true value and thus $\beta_W^{adj} \approx 1$ in normalised coordinates, Eq. (A1) simplifies to,

$$(\mu_i')^2 \approx (\varepsilon_i' - \delta_i')^2 \tag{A5}$$

In essence, the central trend line defined by the revised statistical model of Eq. (31) for the reweighted normalised heights ($h_i^*$) is equivalent to GM fitting corresponding to Eq. (22) for the original normalised heights ($h_i'$). The $\chi^2$-minimisation for the two

models are related as,

$$\frac{d}{d\beta_W^{adj}} \left[ \sum_{i=1}^{N} \frac{(h_i^* - \beta_W^{adj} d_i')^2}{1 + (\beta_W^{adj})^2} \right] \equiv \frac{d}{d\beta_W} \left[ \sum_{i=1}^{N} \frac{(h_i' - \beta_W d_i')^2}{1 + \beta_W^2} \right] \tag{A6}$$

where $\beta_W = 1$ for $\lambda = 1$. Using Eq. (A1) and Eq. (A5), the merit function inside the parenthesis of the left-side of Eq. (A6) approximates to,

$$\sum_{i=1}^{N} \frac{(-\delta_i' + \varepsilon_i' - \mu_i')^2}{2} \approx \sum_{i=1}^{N} \frac{(-\delta_i' + \varepsilon_i')^2 + (\mu_i')^2}{2} \approx \sum_{i=1}^{N} (\mu_i')^2 \tag{A7}$$

where again we have assumed that all the error terms are mutually independent and uncorrelated to the measured variables. Comparison of Eq. (A7) with the merit function on the right-hand side of Eq. (A6) allow an approximate analytic solution of $s_{\mu'}$,

$$s_{\mu'} = Var\left( \sqrt{\frac{(h_i' - d_i')^2}{2}} \right) \tag{A8}$$

From Eq. (A2), Eq. (A3) and Eq. (A4), the variance of the reweighted normalised heights can be evaluated in terms of the

original measurements as,

$$\nu^2 = Var(h_i^*) = 1 - Var\left( \sqrt{\frac{(h_i' - d_i')^2}{2}} \right) \tag{A9}$$

*Author contributions.* ES developed the method, carried out all data analysis and wrote the manuscript. The idea was suggested by AK, who also supervised this work. TU prepared the script for reading the raw data files from the radar, and supervises the doctoral thesis of ES. IV co-supervises the doctoral thesis of ES and made suggestions. BK provided the lidar data. ML supported the MR operation at the Sodankylä.

All the authors contributed to proof reading the manuscript.

*Competing interests.* no competing interests are present

*Acknowledgements.* ES thanks the University of Oulu's Kvantum Institute for their support. IV acknowledges support from the Academy of Finland, project 301542. BK acknowledges support of the German Research Foundation (DFG), research unit Multiscale Dynamics of





Gravity Waves (MS-GWaves) grant RA 1400/6-1. ML acknowledges support from the Science and Technologies Facilities Council (STFC)
grant ST/S000429/1.



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
