# Peer review of "Improved method of estimating temperatures at meteor peak heights"

_Atmospheric Measurement Techniques, 2020_

## Referee Comment (RC1) · Anonymous Referee #1 · 10 Dec 2020

The manuscript "Improved method of estimating temperature at meteor peak heights" by Sarkar et al. deals with a long-debated issue of an accurate temperature estimation method using meteor echoes. They have statistically scrutinized the issue by taking into account the nature of detected echoes, and shown for the first time a method which could estimate the 'log(decay-time) vs height' slope without additional information other than the meteor echoes themselves. Although I think that the method is still crude and has a lot to be improved, their trial can be much appreciated. I therefore would recommend the publication of the manuscript once the following remarks are addressed accordingly.
* * *
General Comments
* * *
I should say that the manuscript is not very easy to follow mostly because of the use of lots of variables, concepts, and definitions. This might be indispensable due to the nature of the manuscript full of mathematical treatment, and also due to my limited skill in mathematics. My most concern regarding the present work is, however, the description in the appendix. It is the most important part of the manuscript dealing with the key correction term, $\mu'i$ or $S\mu'$. I recommend that the authors include it to the main body of the manuscript if no word limit exists. The derivation of the first equation A1 is not clear to me. Eq31 does not imply A1 to me maybe because of my mathematical skill. The mathematical and physical meaning of the final result A8 needs to be more explained. This is the key part of the manuscript. I should admit that I don't fully understand the meaning of A8 although I can half guess the meaning from its simple relation and the similarity with a GM formula of Eq28. A schematic explanation using a plot like Figure 1 would be beneficial for readers.

Further, I am not sure if the final results of $\lambda$=1.76 and $\beta$=0.75 can be regarded as "close to 1" (line 520), which is a necessary condition to derive the analytic solution of $S\mu'$ in the appendix. Since the GM solution needs to be a good approximate of the final slope and the correction should be small enough, an attempt to reduce unwanted error sources, which are referred to as "equation error" in the manuscript, would be necessary and justified although the authors avoid such rejection as "arbitrary". The effect by such equation error sources may not be so severe that the GM approximation fails. The proposed method apparently works well, at least for the meteor data used here. But it is only justified when another temperature data such as lidar data to compare is available. Please note that I don't mean to criticize the authors' effort. Since the idea of introducing a correction term is superb, I would be pleased if I could contribute to the proposed method. In the following paragraphs I will show my idea on "natural variances"/"equation error" in meteor echo observations, because of which I think that an adequate data selection/rejection would be justified prior to applying the proposed technique.

In order to see the nature of the observed meteor echo distribution I over-plotted the GM and final slopes in Figure 1b (a careful replica of Fig1a on Power Point). A close inspection will tell us that the GM solution (red) clearly overestimates a slope which the core distribution inside the 0.4 and 0.8 contour lines indicates while the final slope (black) follows in a better way the core distribution (this is good of course), especially the 0.8 contour area. This implies that the use of core distribution could be a good way to get a better GM solution although the GM solution does not explicitly show up in the equations (35) and (37). The values of final $\lambda$ and $\beta$ would be closer to 1 than the proposed use-every-thing method. This will be because the most annoying regions, the areas surrounded by the orange dotted lines, can be tactfully avoided. The usefulness of this approach should be tested experimentally.

[Figure]

The authors think that the distribution in these orange areas are affected by natural variations and should not be rejected in an arbitrary way. It will be a sincere attitude, but I believe through my long experience in meteor echo study that such distribution is mostly a result of the limitation of observation techniques. For the upper orange distribution, any magnetic field effect may exist, but observation made with a long radio wavelength (2.4 MHz) indicates that ambipolar diffusion shows an exponential increase, at least, up to 110 km without restricted as seen in Figure 7 of Tsutsumi and Aso (2005). The apparent clip seen at around 95 km in the upper part of Figure 1 of the present study is a manifestation of the limited sampling speed and the height ceiling effect (Lee et al., 2018) for a VHF system. Such effect is also seen in the 2.4 MHz observation, but at around 105-110 km. On the other hand the lower orange region is somewhat complicated. Although a chemical effect

may exist, the signal-to-noise ratio of echoes in the region can be responsible, at least partly. The SNR in this region is obviously lower than that in the higher region (please check this with your data) leading to noisier estimate of ambipolar diffusion and mostly apparent offset toward larger diffusion. Because of these reasons an adequate rejection will be justified for a better first estimate of GM solution. There would be no need to pay respect to system-dependent and non-natural error sources. Asymmetric error sources could be handled with Sµ, but a better GM estimate should be tried for a real independent slope estimate without any external temperature information.

Because of the above mentioned height dependent error sources (and more perhaps) the assumption made in the following part of the manuscript is weak as the authors have already realized.

  Lines 140-142
  Lines 229-230 and Eq23
  Line 520
  Line 530

After applying an adequate rejection criterion the assumption can be more acceptable and the proposed fitting method will be more applicable and yield a reliable slope estimate.
* * *
Individual comments
* * *
Figures 2, 3 and 6 X-axis: The dates at the tick marks seem shifted by 15 days.

Line 110: The daily echo numbers around 2000-4000 seem somewhat smaller than expected as a SKiYMET system although I know the number is reduced around spring equinox because of the tilted earth's axis. Some tuning on the radar system may improve the number; antenna tuning, impedance matching.... This is just a comment not necessary to be addressed in the revision.

Figure 3 (a): The figure and caption do not correspond with each other. Figure 3 (a) shows estimated temperatures including those by the lidar (black), and the caption indicates what are plotted is "temperature offset".

Lines 276-277 "the standard errors in these temperatures, which is on average 19 K":
Is this a value estimated using one season of the CORAL lidar data?

Lines 287-289: The use of high contour density area seems worth trying. Or the use of high SNR echoes and/or small zenith angle echoes can be another choice for a better slope estimate because of their less height and decay time estimation errors.

Line 299: Some words are fallen out?

Line 301 "only": Is this necessary?

Lines 317-318: Since it is the decay time that is mostly affected in the lower and upper distribution rather than height as seen in Fig 1, it seems natural to use a correction term to decrease the effective variances of $d_i$ instead of to increase that of $h_i$. Is such an approach possible? I presume it will give an equivalent result.

Lines 331-332: A strict mathematical treatment of $\mu_i$ is beyond my understanding. But is such a practical approach that a constant value $S_{\mu}{}'$ represents the whole equation error mathematically acceptable? Or simply practical? Maybe a meaningless question...

Line 342: Two of the four $\nu$ redundant?

Line 353: What are the bars over $h$ and $d$?

Line 358: Could you explain more about "a priori knowledge"? The knowledge of $\lambda$ being larger or smaller than 1 as well as closer to or far from 1 seems important to decide what model to be used. If so, is a certain amount of shift from 1 (always positive or negative) necessary to apply a model to the data? Always approach from a fixed side to the final solution? This may be another key point of the present method (or I totally misunderstand it).

Line 366-267: Does this mean that the resampling was made 20000 times for every 24 hr?

Line 415: "can estimated" > "can be estimated"

Appendix: More detailed description and mathematical explanation/insight of A8 are wanted as mentioned in the general comments since this is the most important part of the self-consistent slope estimation.

---

## Author Comment (AC1) · 22 Dec 2020

**Discussion**

Emranul Sarkar et al.

December 22, 2020

**1 Data selection process**

Following reviewer's recommendation, we will make adequate data selection in the revised manuscript.

Data will be selected as:

- 1. Only data within the contour line 0.4 will be selected to remove bad data in the outer contour area. This will roughly correspond to the altitude range 85–95 km.
- 2. Zenith angle selection will be now within  $50^{\circ}$  (previously it was  $60^{\circ}$ )

All figures, table and numerical values will be updated accordingly, and shown in Appendix A.

**2 Derivation in the Appendix**

We apologise for few typos (line 513 and 526) in the Appendix. The statement such that the expectation value of  $\mu'$ ,  $or < \mu' > \neq 0$  should be such that the expectation value of  $\mu'$ ,  $or < \mu' >= 0$  (this is from Carroll and Rupper(1996)). The derivation is repeated below with additional explanation:

1. Motivation:

The GM solution in the normalised coordinate is given by -

$$\beta_W = \frac{\sqrt{s_{h'}}}{\sqrt{s_{d'}}} = 1 \tag{1}$$

because  $s_{h'} = s_{h'} = 1$  due to the normalisation with Eq. (18) (manuscript). We found out that this solution is over-estimated, and thus positively biased. Mathematically, this means we can correct for this bias by multiplying the h' with a scaling factor  $\nu$ , such that  $\nu \leq 1$  (this will reduce the numerator in the above equation).

In other words, this reweighted normalised variable is now redefined as  $\nu h'$  or simply  $h^*$ . But this is just a mathematical treatment, the observed value of h' will be the same observed value. The difference between the variance of our hypothetical variable  $h^*$  and the real observable h' is some constant, say  $s_{\mu'}$  (where  $\mu'$  is  $h' - h^*$ ),

$$s_{h^*} = s_{h'} - s_{\mu'} = 1 - s_{\mu'} \tag{2}$$

2. But Eq. (2) above also implies,

$$h' = h^* + \mu' \tag{3}$$

In the manuscript, we have defined h' in line-139 as (but now in normalised coordinate, hence all with *prime* subscript, and  $\alpha = 0$ ),

$$h'_{i} = \beta_{W}\xi'_{i} + \varepsilon'_{i} = \beta_{W}(d'_{i} - \delta'_{i}) + \varepsilon'_{i}$$

$$\tag{4}$$

Substitute (3) in (4) to get,

$$h_i^* + \mu_i' = \beta_W^{adj} \xi_i' + \varepsilon_i' = \beta_W^{adj} (d_i' - \delta_i') + \varepsilon_i'$$
(5)

Note that now we write the normalised slope as  $\beta_W^{adj}$ , where  $\beta_W^{adj} \leq \beta_W$  (of course,  $\beta_W = 1$ ). Equation (5) above is Eq. (A1) in the manuscript (line-512).

3. Solution: Assume  $\beta_W^{adj} \approx \beta_W$

Write Eq. (5) as -

$$h_i^* - \beta_W^{adj} d_i' + \mu_i' = -\beta_W^{adj} \delta_i' + \varepsilon_i' \tag{6}$$

And so,

$$(\mu_i')^2 \approx (\varepsilon_i' - \delta_i')^2 \tag{7}$$

Equate the residual function of GM solution (line 251) with its perturbed value as (see Fig. 1 for a bit of visual understanding)-

$$\sum_{i=1}^{N} \frac{(h_i^* - \beta_W^{adj} d_i')^2}{1 + (\beta_W^{adj})^2} \approx \sum_{i=1}^{N} \frac{(h_i' - \beta_W d_i')^2}{1 + \beta_W^2}$$
(8)

Substitute Eq. (6) and Eq.(7) in Eq. (8), and with  $\beta_W^{adj} \approx \beta_W = 1$ , the left-hand side of Eq.(8)

$$\sum_{i=1}^{N} \frac{(-\delta_i' + \varepsilon_i' - \mu_i')^2}{2} \approx \sum_{i=1}^{N} \frac{(-\delta_i' + \varepsilon_i')^2 + (\mu_i')^2}{2} \approx \sum_{i=1}^{N} (\mu_i')^2 \tag{9}$$

Or Eq. (8) is now,

$$\sum_{i=1}^{N} (\mu'_i)^2 = \sum_{i=1}^{N} \frac{(h'_i - d'_i)^2}{2}$$
(10)

$$\frac{1}{N}\sum_{i=1}^{N} \mid \mu_{i}^{'} \mid = \frac{1}{N}\sum_{i=1}^{N}\sqrt{\frac{(h_{i}^{'}-d_{i}^{'})^{2}}{2}}$$
(11)

Bienaymé formula1 states that for uncorrelated random variable  $Y_i$ :

$$Var\left(\sum_{i=1}^{N} Y_i\right) = \sum_{i=1}^{N} Var(Y_i)$$
(12)

Or,

<sup>1https://en.wikipedia.org/wiki/Variance

$$Var\left(mean(Y_i)\right) = Var\left(\frac{1}{N}\sum_{i=1}^{N}Y_i\right) = \frac{1}{N^2}Var\left(\sum_{i=1}^{N}Y_i\right) = \frac{1}{N}Var(Y_i)$$
(13)

Apply the formula in Eq. (13) to Eq. (11)

$$Var\left(\frac{1}{N}\sum_{i=1}^{N}|\mu_{i}'|\right) = Var\left(\frac{1}{N}\sum_{i=1}^{N}\sqrt{\frac{(h_{i}'-d_{i}')^{2}}{2}}\right)$$
(14)

Or,

$$Var\left(\mu_{i}^{'}\right) = Var\left(\sqrt{\frac{(h_{i}^{'}-d_{i}^{'})^{2}}{2}}\right)$$
(15)

From Eq. (2), we therefore have,

$$\nu^2 = s_{h^*} = 1 - Var\left(\sqrt{\frac{(h'_i - d'_i)^2}{2}}\right)$$
(16)

An alternative, and perhaps much straight-forward way to go from Eq. (11) to Eq. (15) is to utilise the property of half-normal distribution2. This states that if Y follows an ordinary normal distribution, with mean 0 and variance  $\sigma^2$ , then X = |Y| follows a half-normal distribution, such that the variance of X is given by-

$$Var(X_i) = \sigma^2 (1 - \frac{2}{\pi}) \tag{17}$$

And their expectation value is related as-

$$E(X) = \sigma \frac{\sqrt{2}}{\pi} \tag{18}$$

Since  $\mu_i$  (left-hand side) and the residuals from GM solution (right-hand side) in Eq. (11) are assumed to have zero mean and normally distributed, their absolute value follows half-normal distribution. Applying the formula in Eq. (18) to Eq. (11) directly lead to Eq. (15).

**Note:**

• How is Eq. (5) related to the statistical model in line-305 of the manuscript?

We have originally defined the normalised variable  $h'_i$  (observable) in terms of hypothetical true value  $\eta'_i$  in line 139, for which the normalised slope is  $\beta_W$ :

$$h_{i}^{'} = \eta_{i}^{'} + \varepsilon_{i}^{'} \tag{19}$$

According to the model with 'equation error' in line-305, we need to replace  $\eta'_i$  by  $\eta'_i - \mu_i$  so that the variance of observed height  $h'_i$  is reduced to that of  $h^*$ .

$$h_i^* = \eta_i' - \mu_i + \varepsilon_i' \tag{20}$$

<sup>2https://en.wikipedia.org/wiki/Half-normal\_distribution

Following reviewer comment, we have realised that this information above is quite redundant since Eq. (5) didn't specifically require this. In the revised manuscript, we will use adequate data selection and hence such extended discussion on 'natural variation' or 'equation error' (Sect. 3.2.2) is no longer necessary. Instead, we will simply refer to  $s_{\mu}$  as asymmetric error.

• Numerical validation of the equality in Eq. (8) above:

For the date 14 Nov 2015, the RHS of Eq. (8) is -

$$\sum_{i=1}^{N} \frac{(h'_i - d'_i)^2}{2} \approx 85.87 \tag{21}$$

In the current manuscript we have  $\beta_W^{adj} \approx 0.75$  and  $\nu \approx 0.87$ , and so the LHS of Eq. (8) is -

$$\sum_{i=1}^{N} \frac{(h_i^* - \beta_W^{adj} d_i')^2}{1 + (\beta_W^{adj})^2} \approx 75.50$$
(22)

In the revised manuscript (with adequate bad data removal) we have  $\beta_W^{adj} \approx 0.95$  and  $\nu \approx 0.96$  (see Fig. A.8), and now the LHS of Eq. (8) is -

$$\sum_{i=1}^{N} \frac{(h_i^* - \beta_W^{adj} d_i')^2}{1 + (\beta_W^{adj})^2} \approx 82.35$$
(23)

- This manuscript do not make the default assumption that  $\lambda = 1$  in the normalised coordinate is indeed the fundamental property of this distribution, and hence the motivation to develop a solution that allows for small variation in  $\lambda$  from 1. This is because the exact value of  $\lambda$  depends on the data selection process (and also probably on the instrument, but this study is just with one radar).
- In the revised manuscript we will state the 3 possible cases:
  - 1.  $\lambda = 1$ . The GM solution is valid.
  - 2.  $\lambda \gtrsim 1$ . Eq. (37) is valid.
  - 3.  $\lambda \lesssim 1$ . Eq. (37) is valid but with h and d interchanged.

**3 Specific**

**3.1 This implies that the core distribution could be a good way to get a better GM solution although the GM solution do not explicitly show up in Eq. (35) and Eq. (37)**

Eq. (35) and Eq. (37) are indeed showing this convergence. Note that these equations are given in normalised coordinates  $(d'_i \text{ and } h'_i)$ . Now, if we look closely in Fig. 1b (manuscript) or Fig. A.1b here, we see that as we approach the core of this distribution, both  $d'_i$  and  $h'_i$  gets smaller and converges to zero-point. This means both Eq. (35) and Eq. (37) converges to 1 near the peak. This is demonstrated in Fig. 1 (below) where we have estimated the slope for GM solution and with Eq. (37) using 6 different data selection criteria. The stability of the solution from Eq. (37) arises due to the correction term  $\nu$  which is a constant that depends on the input data via Eq. (16) above (or, Eq. (35) in the manuscript).

---

## Author Comment (AC2) · 7 Jan 2021

The value of B_W(adj) in Fig. 1e (Con,0.4;Ze<50) is 0.945.

———————————————

---

## Referee Comment (RC2) · Anonymous Referee #2 · 25 Jan 2021

Review Manuscript: amt-2020-333.pdf
Title: Improved method of estimating temperatures at meteor peak heights.

The authors apply Errors-in-Variables (EIV) modeling to the temperature-gradient (Hocking et al. [1997] and Hocking [1999]) method of estimating temperatures at meteor peak heights. The application of EIV is shown to improve temperature estimation without the ad-hoc calibration previously used. The authors recognize that the total variance (geophysical and parameter estimation error) needs to be used as the EIV model does not distinguish between the two sources of variability in the model equation.

I am surprised that no-one has done this previously. Thorsen et al [1997 Radio Science, V32, N2, pp707-726] applied total least squares (TLS – Van Huffel and Vandewalle [1991]), an equivalent technique to EIV (see Editorial Computational Statistics & Data Analysis 52 (2007) 1076-1079), to the estimation of the mean wind field in the middle atmosphere using estimates of radial velocity and angle of arrival of echoes in a similar manner to how MR data is analyzed. They came to the same conclusion that the model error included the geophysical variability as well as the parameter estimation error and that the geophysical variability was the larger contributor to the model error.

Line 23:
If the electron line density of the trail is less than $2.4 \times 10^{14}$ electrons $m^{-1}$ the trail is called 'underdense'….

The value of the electron line density that marks the transition between underdense and overdense meteor trails is frequency dependent.

The authors should either state that: "for frequency 36.9 MHz or for the Sodankylä meteor radar the electron line density of the trail is less than $2.4 \times 10^{14}$ electrons $m^{-1}$ the trail is called 'underdense'…"

Section 2:
It is not clear in the paper that the author is applying their analysis on underdense echoes only, rather than the underdense and overdense echoes that the SKiYMet system detects as valid meteors. Author only talks about restricting detections due to large radial velocity. Does this analysis include overdense echoes? If so, how does the author justify using an underdense model on an overdense echo?

Line 163
To this date, no such attempt has been made to assess these error variances in MR data.

Although this may be true specifically for MR data, error variances can be calculated theoretically, see Zrnić [1977], Doviak and Zrnić [1993], Woodman and Hagfors [1969] (referenced in Thorsen et. al. [1997]) for comparable calculations. Just because the effort has not been made, does not mean that it can't be made.

However, since the geophysical variability is likely to dominate over the parameter estimation errors this lack is potentially moot.  Thorsen et al [1997] performed the comparison between the parameter estimation error and the geophysical variability and found that the geophysical variability dominated at all heights.

Line 290
However, temperatures estimated with any such arbitrary choice of data rejection criteria will lack consistency.

Is this will or do lack consistency?  Was this tested or is this statement an assumption?  Actually, I'm not sure I understand what is meant by "will lack consistency."

---

## Author Response (AR1)

**Author Comments**

Emranul Sarkar et al.

March 22, 2021

We thank the reviewers for useful comments. We carefully considered all of them and revise the manuscript accordingly. We delete Appendix and everything dealing with the correction term $s'_\mu$. Instead, in the revised manuscript we apply removal of the data outside certain contour lines in the height-log (1/tau) distribution. The contour line selection and improved temperature estimate are illustrated in attached Figures 1-7. Correspondingly, the manuscript in section 3.2.2 and below is essentially rewritten. Large fragments of the former text are removed in this part, such that a number of specific comments are addressed due to the text removal. Below are point-to-point replies (in *blue*) to the reviewers' questions.

Authors changes, the marked-up version (the changes made in *blue* and the lines removed in *red*) and the revised paper are attached below.

**1    RC1**

**I should say that the manuscript is not very easy to follow mostly because of the use of lots of variables, concepts, and definitions. This might be indispensable due to the nature of the manuscript full of mathematical treatment, and also due to my limited skill in mathematics.**

(1) Part of this paper is a review of earlier studies, and we mostly cite variables, concepts, and definitions from these papers.

**My most concern regarding the present work is, however, the description in the appendix. It is the most important part of the manuscript dealing with the key correction term, $s_{\mu'}$. I recommend that the authors include it to the main body of the manuscript if no word limit exists. The derivation of the first equation A1 is not clear to me. Eq31 does not imply A1 to me maybe because of my mathematical skill. The mathematical and physical meaning of the final result A8 needs to be more explained. This is the key part of the manuscript. I should admit that I don't fully understand the meaning of A8 although I can half guess the meaning from its simple relation and the similarity with a GM formula of Eq28. A schematic explanation using a plot like Figure 1 would be beneficial for readers.**

(2) This part is entirely removed in the revised paper, and we use the contour selection method instead for a better GM solution.

**In order to see the nature of the observed meteor echo distribution I over-plotted the GM and final slopes in Figure 1b (a careful replica of Fig1a on Power Point). A close inspection will tell us that the GM solution (red) clearly overestimates a slope which the core distribution inside the 0.4 and 0.8 contour lines indicates while the final slope (black) follows in a better way the core distribution (this is good of course), especially the 0.8 contour area. This implies that the use of core distribution could be a good way to get a better GM solution although the GM solution does not explicitly show up in**

the equations (35) and (37). The values of final $\lambda$ and $\beta$ would be closer to 1 than the proposed use-every-thing method. This will be because the most annoying regions, the areas surrounded by the orange dotted lines, can be tactfully avoided. The usefulness of this approach should be tested experimentally.

(3) We follow these recommendations and consider the contour removal method. Corresponding changes are made throughout the manuscript. Updated figures and table are shown below.

The authors think that the distribution in these orange areas are affected by natural variations and should not be rejected in an arbitrary way. It will be a sincere attitude, but I believe through my long experience in meteor echo study that such distribution is mostly a result of the limitation of observation techniques. For the upper orange distribution, any magnetic field effect may exist, but observation made with a long radio wavelength (2.4 MHz) indicates that ambipolar diffusion shows an exponential increase, at least, up to 110 km without restricted as seen in Figure 7 of Tsutsumi and Aso (2005). The apparent clip seen at around 95 km in the upper part of Figure 1 of the present study is a manifestation of the limited sampling speed and the height ceiling effect (Lee et al., 2018) for a VHF system. Such effect is also seen in the 2.4 MHz observation, but at around 105-110 km. On the other hand the lower orange region is somewhat complicated. Although a chemical effect may exist, the signal-to-noise ratio of echoes in the region can be responsible, at least partly. The SNR in this region is obviously lower than that in the higher region (please check this with your data) leading to noisier estimate of ambipolar diffusion and mostly apparent offset toward larger diffusion. Because of these reasons an adequate rejection will be justified for a better first estimate of GM solution. There would be no need to pay respect to system-dependent and non-natural error sources. Asymmetric error sources could be handled with $s_\mu$, but a better GM estimate should be tried for a real independent slope estimate without any external temperature information. Because of the above mentioned height dependent error sources (and more perhaps) the assumption made in the following part of the manuscript is weak as the authors have already realized.

Lines 140-142,

Lines 229-230 and Eq23,

Line 520,

Line 530; After applying an adequate rejection criterion the assumption can be more acceptable and the proposed fitting method will be more applicable and yield a reliable slope estimate.

(4) Corresponding changes are made in the revised manuscript. See (2) and (3) above.

Figures 2, 3 and 6 X-axis: The dates at the tick marks seem shifted by 15 days.

Line 110: The daily echo numbers around 2000-4000 seem somewhat smaller than expected as a SKiYMET system although I know the number is reduced around spring equinox because of the tilted earth's axis. Some tuning on the radar system may improve the number; antenna tuning, impedance matching. This is just a comment not necessary to be addressed in the revision.

Figure 3 (a): The figure and caption do not correspond with each other. Figure 3 (a) shows estimated temperatures including those by the lidar (black), and the caption

indicates what are plotted is 'temperature offset'.

**Line 299: Some words are fallen out?**

**Line 301 'only': Is this necessary?**

**Line 415: 'can estimated' >'can be estimated'**

(5) These are fixed in the revised paper. The word 'only' can be deleted without any loss of meaning.

**Lines 276-277 'the standard errors in these temperatures, which is on average 19 K': Is this a value estimated using one season of the CORAL lidar data?**

(6) No, this was estimated independently using bootstrap analysis. For GM slope, the standard error can be obtained using an analytic formula (Vicente de Julian-Ortiz, J., Lionello Pogliani, and Emili Besalu. 'Two-variable linear regression: modeling with orthogonal least-squares analysis.' Journal of chemical education 87.9 (2010): 994-995.). Alternatively, the standard error in GM slope can be estimated from the standard error in two OLS slopes.

**Lines 287-289: The use of high contour density area seems worth trying. Or the use of high SNR echoes and/or small zenith angle echoes can be another choice for a better slope estimate because of their less height and decay time estimation errors.**

**Lines 317-318: Since it is the decay time that is mostly affected in the lower and upper distribution rather than height as seen in Fig 1, it seems natural to use a correction term to decrease the effective variances of $d_i$ instead of to increase that of $h_i$. Is such an approach possible? I presume it will give an equivalent result.**

**Lines 331-332: A strict mathematical treatment of $\mu_i$ is beyond my understanding. But is such a practical approach that a constant value $s_{\mu'}$ represents the whole equation error mathematically acceptable? Or simply practical? Maybe a meaningless question...**

**Line 342: Two of the four $\nu$ redundant?**

**Line 353: What are the bars over h and d?** 'Mean value of measured $d_i$ and $h_i$'.

**Line 358: Could you explain more about 'a priori knowledge'? The knowledge of $\lambda$ being larger or smaller than 1 as well as closer to or far from 1 seems important to decide what model to be used. If so, is a certain amount of shift from 1 (always positive or negative) necessary to apply a model to the data? Always approach from a fixed side to the final solution? This may be another key point of the present method (or I totally misunderstand it).**

**Line 366-267: Does this mean that the resampling was made 20000 times for every 24 hr?** 'Yes'.

**Appendix: More detailed description and mathematical explanation/insight of A8 are wanted as mentioned in the general comments since this is the most important part of the self-consistent slope estimation.**

(7) See (2) and (3) above.

**2 RC2**

**Line 23: If the electron line density of the trail is less than $2.4 \times 10^{14}$ electrons m-1 the trail is called 'underdense'.... The value of the electron line density that marks the transition between underdense and overdense meteor trails is frequency dependent. The authors should either state that:** *for frequency 36.9 MHz or for the Sodankylä meteor radar the electron line density of the trail is less than $2.4 \times 10^{14}$ electrons $m^{-1}$ the trail is called 'underdense'...*

(8) On definition, classification of meteor trails does depend only on the line electron density, and does not depend on the radar frequency (e.g., Bronshten, 1983, page 219-220).

**It is not clear in the paper that the author is applying their analysis on underdense echoes only, rather than the underdense and overdense echoes that the SKiYMet system detects as valid meteors. Author only talks about restricting detections due to large radial velocity. Does this analysis include overdense echoes? If so, how does the author justify using an underdense model on an overdense echo?**

(9) The radar signal processing does not separate under- and over-dense trails. However, most of meteors detected by SKiYMET ($> 95\%$) are underdense (Hocking et al., 2001). The percentage of overdense trails may be larger during some meteors trails such as Geminids or Quadrantids (Kozlovsky et al., 2016). This leads to underestimated temperature during peaks of these showers on 13-14 December and 3 January, respectively. This issue is discussed in the revised paper.

**Line 163:** *To this date, no such attempt has been made to assess these error variances in MR data.*

(10) This statement was made in the context of SCT calibration. We will rephrase this in the revised paper.

**Although this may be true specifically for MR data, error variances can be calculated theoretically, see Zrnić [1977], Doviak and Zrnić [1993], Woodman and Hagfors [1969] (referenced in Thorsen et. al. [1997]) for comparable calculations. Just because the effort has not been made, does not mean that it can't be made. However, since the geophysical variability is likely to dominate over the parameter estimation errors this lack is potentially moot. Thorsen et al [1997] performed the comparison between the parameter estimation error and the geophysical variability and found that the geophysical variability dominated at all heights.**

(11) This will be emphasised in the revised paper with additional citation.

**Line 290: However, temperatures estimated with any such arbitrary choice of data rejection criteria will lack consistency. Is this will or do lack consistency? Was this tested or is this statement an assumption? Actually, I am not sure I understand what is meant by will lack consistency.**

(12) We delete this line in the revised paper.

**3 Updated Figures and Table**

[Figure]

Figure 1: (a) Typical scatter plot of $log_{10}(1/\tau)$ and height. The lines correspond to best fit models using different regression methods described in the text. The *green* and *blue* line corresponds to 'ordinary least-squares method (OLS)' with $log_{10}(1/\tau)$ and height as independent variable respectively. The *red* line correspond to the geometric mean (GM) of $\beta_{OLS}^d$ and $\beta_{OLS}^h$. (b) The bivariate distribution of the data. The measured height and $log_{10}(1/\tau)$ are converted to dimension free coordinates using Eq. (18). The relative density contours are obtained by counting the number of detections in a circle of unit area relative to the density at the height of peak meteor occurrences at the center.

[Figure]

Figure 2: (a) Temperature gradient model derived from MSIS90. (b) Peak meteor heights for the data used in this work, and (c) the daily meteor detection for zenith angle less than $50°$ and velocity in the range $\pm 100$ m/s.

[Figure]

Figure 3: (a) Temperature estimated in OLS method using $log_{10}(1/\tau)$ (*green*) and height (*blue*) as independent variable. Also, showing (*red*) the temperatures obtained using the geometric mean (GM) fitting. (b) The offset between the lidar ($T_{lidar}$) temperatures and the estimated MR temperatures ($T_{MR}$) using OLS fitting and GM fitting (without contour selection).

[Figure]

Figure 4: GM solution at different contour levels in (a) original coordinate and in (b) normalised coordinate. The vertical dashed lines correspond to the average value of $\lambda$ obtained from SCT calibration at contour level 0.2 and 0.4 respectively for winter 2015–2016.

[Figure]

Figure 5: Comparison of the bias-corrected MR temperatures with lidar data for the winter 2015–2016. The *solid* line corresponds to the temperature estimated using the GM solution at contour level 0.4. The *dashed* line corresponds to the SCT calibrated temperatures using the colocated lidar measurements. The OLS estimates are obtained with $log_{10}(1/\tau)$ as independent variable. The errors in lidar temperatures are 5–10 K and the standard error (grey shade) of the temperature from EIV analysis is on average 19 K. The differences between lidar and MR temperatures are presented in Fig. 6.

[Figure]

Figure 6: Difference between MR temperatures and lidar data for (a) GM solution at contour level 0.4 and (b) SCT calibration applied to OLS estimate of $\beta_{OLS}^d$. MR temperatures are shown in Fig. 5.

Table 1: Average value of the (square root of) error variances and their normalised ratio for SGO's meteor radar obtained by SCT method for winter 2015–2016. The average value of the error variances in normalised height and $log_{10}(1/\tau)$, $s_{\varepsilon'}$ and $s_{\delta'}$, are given along with the average value of $\lambda$ from SCT calibration ($\lambda_{eff}^{OLS}$) at contour levels 0, 0.2 and 0.4.

| Contour: | 0 | 0.2 | 0.4 |
|---|---|---|---|
| $\triangle(height)/km$ | 5.0 | 3.0 | 2.2 |
| $\triangle(log_{10}(1/\tau))/s^{-1}$ | 0.18 | 0.14 | 0.11 |
| $< s_{\varepsilon'} >$ | 0.62 | 0.44 | 0.38 |
| $< s_{\delta'} >$ | 0.37 | 0.31 | 0.30 |
| $< \lambda_{eff}^{OLS} >$ | 1.67 | 1.43 | 1.25 |

[Figure]

Figure 7: (a) Improved temperature estimation using GM solution (red) as compared to OLS estimates (blue and green) at contour level 0.4. (b) Reduced mean offset between MR and lidar temperature for GM slope estimate (red) as compared to OLS estimate (green and blue).

**4 Author's changes in the Marked-up version**

1. Parts of 'Abstract' is rewritten to emphasise the biasing effect that can arise due the observational limitation of meteor radar. Data selection process based on contour levels is included.

2. The *black* line corresponding to the correction term $s'_\mu$ is removed from Fig. 1(a), and corresponding changes are made in the caption (Pg 2).

3. Citation (Bronshten., 1983) included in line 29 to emphasise the relation between electron line density and 'underdense/overdense' trails.

4. Line 110–114: The justification of using underdense model with data from SKIYMET system is explained.

5. 115–117: The data selection is now restricted to elevation angle above 40°. Correspondingly Fig. 2(c) is updated in Pg-5.

6. The word 'offset' is removed from the caption of Fig. 3(a) and this figure is updated accordingly.

7. Line 180–183 removed. Table 1 (pg 11) updated to include the error estimates at various contour levels.

8. Line 205–234: The dominant effect of geophysical variability is discussed. Thorsen et al. (1997) is cited.

9. Section 3.2. Large fragments of the former text are removed in this part, such that a number of specific comments by referees are incorporated. We delete Appendix and everything dealing with the correction term $s'_\mu$. Instead, in the revised manuscript we apply removal of the data outside certain contour lines in the height-log (1/tau) distribution. Fig. (4) and Fig. (7) are added to support the discussion.

10. Section 4. All discussions related to $s'_\mu$ are removed. The biasing effect in GM solution is discussed within the context of the parameter $\lambda$. the captions in Fig. (4) and Fig. (7) are updated.

11. Line 420–435: Bootstrap analysis is removed. The standard error is obtained using an analytic formula (Vicente de Julian-Ortiz, J., Lionello Pogliani, and Emili Besalu. 'Two-variable linear regression: modeling with orthogonal least-squares analysis.' Journal of chemical education 87.9 (2010): 994-995.)

12. Line 545–575: Discussion of Pressure-method and citation of Lee et al. (2018) are removed as these are not directly relevant to the main objective of this paper.

13. Pg 591–607: The summary section in updated according to the changes made in the revised paper.

14. A marked-up version showing the changes made in the paper and the revised paper are attached below.

[revised manuscript text omitted]

---

## Author Response (AR2)

- Changes are made in the first column of Table 1 following referee's comment.
- Minor grammar correction in the caption of Fig. 4.